# How Many Van Goghs Does It Take to Van Gogh? Finding the Imitation Threshold

## Abstract

Text-to-image models are trained using large datasets collected by scraping image-text pairs from the internet. These datasets often include private, copyrighted, and licensed material. Training models on such datasets enables them to generate images with such content, which might violate copyright laws and individuals privacy. This phenomenon is termed *imitation* – generation of images with recognizable similarity to training images. In this work we study the relationship between a concept's frequency in a dataset and the ability of a model to imitate it. We seek to determine the point at which a model was trained on enough instances to imitate a concept – the *imitation threshold*. We posit this question as a new problem: **F**inding the **I**mitation **T**hreshold (FIT) and propose an efficient approach that estimates the imitation threshold without incurring the colossal cost of training multiple models from scratch. We experiment with two domains – human faces and art styles for which we create three datasets, and evaluate three text-to-image models which were trained on two pre-training datasets. Our results reveal that the *imitation threshold* of these models is in the range of 200-600 images, depending on the domain and the model. The *imitation threshold* can provide an empirical basis for copyright violation claims and acts as a guiding principle for providers of text-to-image models that aim to comply with copyright and privacy laws.

## 1 Introduction

The progress of multi-modal vision-language models has been phenomenal in recent years [14, 35, 36, 38], much of which can be attributed to the availability of large-scale pretraining datasets like LAION [44]. These datasets consist of semi-curated image-text pairs scraped from Common Crawl, which leads to the inclusion of explicit, copyrighted, and licensed material [4, 9, 17, 20, 56]. Training models on such images may be problematic because text-to-image models can *imitate* — the ability to generate images with recognizable features — concepts from their training data [5, 49]. This behavior has both legal and ethical implications, such as copyright infringements as well as privacy violations of individuals whose images are present in the training data without consent. In fact, a large group of artists sued Stability AI, creators of widely-used text-to-image models, alleging that the company's models generated images that distinctly replicated their artistic styles [42].

Previous work has focused on detecting when generated images imitate training images, and mitigations thereof [5, 48–50]. In particular, researchers found that duplicate images increase the chance of memorization and imitation. However, the relation between a concept's prevalence and the models' ability to imitate it remains unexplored.

In this work, we ask **how many instances of a concept does a model need to be trained on to imitate it?** Establishing such an *imitation threshold* is useful for several reasons. First, it provides an empirical basis for copyright infringements and privacy violations claims [42, 56]. Second, it acts as a guiding principle for text-to-image models providers that want to avoid such violations. Finally, it reveals an interesting connection between training data statistics and model behavior, and the ability of models to efficiently harness training data [6, 53]. We name this problem FIT: **F**inding the **I**mitation **T**hreshold, and provide a schematic overview of this problem in Figure 1.

Submitted to Safe Generative AI Workshop @ NeurIPS 2024. Do not distribute.

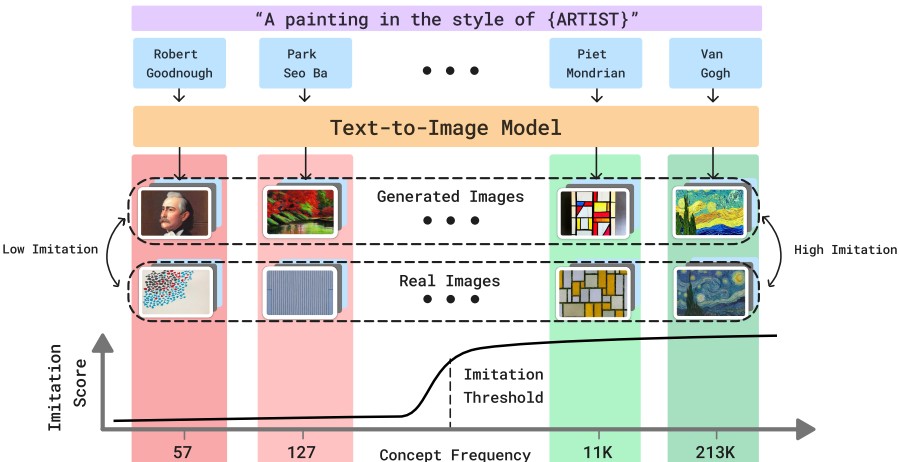

Figure 1: An overview of FIT, where we seek the *imitation threshold* – the point at which a model was exposed to enough instances of a concept that it can reliably imitate it. The figure shows four concepts (e.g., Van Gogh's art style) that have different frequencies in the training data (213K for Van Gogh). As the frequency of a concept's images increases, the ability of the text-to-image model to imitate it increases (e.g. Piet Mondrian and Van Gogh). We propose an efficient approach, MIMETIC$^2$, that estimates the imitation threshold without training models from scratch.

The optimal methodology to measure the imitation threshold requires training multiple models with varying number of images of a concept and measuring the ability of the counterfactual models to imitate it. However, training even one of these models is prohibitively expensive. We propose an alternative approach, **M**easuring **Im**itation Thr**E**shold **T**hrough **I**nstance **C**ount and **C**omparison (MIMETIC$^2$), that estimates the threshold without incurring the cost of training models from scratch. We start by collecting a large set of concepts per domain (e.g., Van Gogh for artistic styles), and use a text-to-image to generate images for each concept. Then, we compute the imitation score of the generated images by comparing them to the training images of the respective concept, and estimate each concept's frequency in the training data. Finally, by sorting the concepts based on frequency we estimate the imitation threshold for that domain using a *change detection* algorithm [21].

Since we operate with observational data, a naive implementation may be confounded by different factors, such as the quality of the imitation scoring model on different groups within the domain, or estimating the training frequencies of concepts (e.g., simple counts of 'Van Gogh' in the captions results in a biased estimate since the artist may be mentioned in the caption without their work). As such, we carefully tailor MIMETIC$^2$ to minimize the impact of such confounders.

Overall, we formalize a new problem – **F**inding **I**mitation **T**hreshold (FIT; §3), and propose a method, MIMETIC$^2$, that efficiently estimates the *imitation threshold* for text-to-image models (§5). We use our method to estimate the imitation threshold for two domains on three datasets, three text-to-image models that were trained on two pretraining datasets (§4). We find the imitation thresholds to range between 200 to 600 images, providing concrete insights on models' imitation abilities (§6).

## 2   Background

**Dataset Issues and Privacy Violations** The advancement in text-to-image capabilities, largely due to big training datasets, is accompanied by concerns about the training on explicit, copyrighted, and licensed material [4] and imitating such content when generating images [9, 17, 20, 56]. For example, Birhane et al. [4] and Thiel [52] found several explicit images in the LAION dataset and Getty Images found that LAION had millions of their copyrighted images [56]. Issues around imitation of training images has especially plagued artists, whose livelihood is threatened [42, 48], as well as individuals whose face has been used without consent to create inappropriate content [2, 17].

**Training Data Statistics and Model Behavior** Pre-training datasets are a core factor for explaining model behavior [12]. Razeghi et al. [37] found that the in-context few-shot performance of language models (LMs) is highly correlated with the frequency of instances in pre-training datasets. Udandarao et al. [53] bolster this finding by demonstrating that the performance of multimodal models on downstream tasks is strongly correlated with a concept's frequency in the pretraining datasets. In

addition, Carlini et al. [6] shows that language models more easily memorize duplicated sequences. We find a similar phenomenon: increasing the number of images of an instance increases the similarity between the generated and training images on average. Crucially, instead of measuring *memorization*, we measure *imitation*, and we use such metric to find the *imitation threshold*.

Samara Joy  Jacquetta Wheeler  James Garner  Stephen King  Johnny Depp

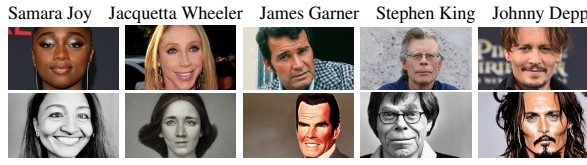
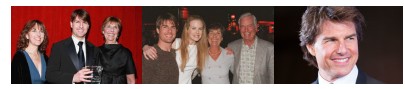

Figure 2(a): Examples of real celebrity images (top) and generated images (bottom) with increasing image counts from left to right (3, 273, 3K, 10K, and 90K, respectively).

Figure 2(b): LAION2B-en images whose caption mentions 'Mary Lee Pfeiffer', the mother of Tom Cruise. She is not always present in the images (the rightmost image has only Tom Cruise).

## 3 Problem Formulation and Overview

Finding the Imitation Threshold (FIT) seeks to find the minimal number of images with some concept a model has to see during training in order to imitate it. FIT's setup involves a training dataset $\mathcal{D} = \{(x_1, y_1), (x_2, y_2), ..., (x_n, y_n)\}$, composed of $n$ (image, caption) pairs. Each concept is part of a domain $\mathcal{O}$, such as art styles. We also assume an indicator $I^j$ that indicates whether a concept $Z^j$ is present in image $x_i$. Each concept $Z^j$ appears $c^j = |\sum_i I^j(x_i)|$ times in the dataset $\mathcal{D}$. Finally, we assume a model $\mathcal{M}$ that is trained on $\mathcal{D}$ as a text-to-image model to predict $x_i$ from $y_i$. The *imitation threshold* is the minimal number of images $c^j$ with some concept $Z^j$ from which the model $\mathcal{M}$ is able to generate images $\mathcal{M}(p^j)$ given some prompt $p^{j\,1}$ where $Z^j$ is recognizable as the concept.

$$\min\left\{k \in \{0, 1, \dots\} : I^j(\mathcal{M}_k(p^j)) = 1\right\}$$

**Optimal Approach.** Finding the *imitation threshold* is a causal question – *if model $\mathcal{M}$ was trained with $k'$ images of concept $Z^j$ instead of $k$, could it generate images with this concept?* The optimal manner of answering this question is a brute force experiment [33]: For each concept $Z^j$, we create a dataset $\mathcal{D}'_k$ where we vary the number of images with such concept in the training data, where $k \in \{0, 1, \dots, n\}$, and train a model $\mathcal{M}'_k$ on each dataset. Once we find a model, $\mathcal{M}'_k$, that is able to generate the concept, but $\mathcal{M}'_{k-1}$ cannot, we deem $k$ as the *imitation threshold* for that concept. *However, due to the extreme costs of training text-to-image models, this optimal approach is impractical (this approach will require training $\mathcal{O}(\log n)$ models).*

**MIMETIC**[2]. We propose an approach that is tractable and estimates the causal effect under certain assumptions. The key idea is to use observational data instead of training a model for different number of images for each concept. Such an approach has been previously used to answer causal questions, inter alia, [25, 29, 33]. Concretely, we collect several different concepts ($Z^j$) belonging to some domain ($\mathcal{O}$) while ensuring that these concepts have varying image frequencies in the training dataset $\mathcal{D}$. Then, we identify the frequency where model $\mathcal{M}$ starts generating images with the concept at that frequency. We term this frequency as the *imitation threshold*.

To evaluate the imitation ability, we build a *concept-score* function $f$ that returns an imitation score $f(X_t, \mathcal{M}_{P^j})$ that measures the imitation of a concept in the generated images using its training data. $X_t := x_1, ..., x_t$ is a set of training images associated with concept $Z^j$. $\mathcal{M}_{P^j} := \mathcal{M}(p^j)_1, \mathcal{M}(p^j)_2, \dots, \mathcal{M}(p^j)_g$ is a set of generated images created using different random seeds and a text prompt that mentions $Z^j$, For a domain $\mathcal{O}$, we collect a set of concepts $Z^1, Z^2, ..., Z^m$ (e.g., a list of artistic styles), estimate each concept's frequency in data $\mathcal{D}$, and measure the imitation score for each concept. Sorting the concepts based on their frequencies in the dataset, and using a standard *change detection* algorithm on the imitation scores, gives us the imitation threshold for that domain. We provide the implementation details in Section 5.

**Assumptions.** Our approach to compute the imitation threshold makes an assumption about distribution invariance in order to make the problem computationally tractable. This assumption is a standard practice when answering causal questions using observational data [33]. This assumption posits an invariance in the image distribution of each concept. Under this assumption, measuring the imitation score of a concept $Z^i$ with a counterfactual model trained with $k'$ images of $Z^i$ is

---

[1]Prompts $p^j$ are usually different from the captions in the training data, $y_i$.

equivalent to measuring the imitation of another concept $Z^j$ that currently has $k'$ images in the already trained model. This helps us answer the causal question FIT seeks without training multiple models. And similar to other sample complexity works [53, 54], we also assume each image of a concept contributes equally to its learning.

## 4  Experimental Setup

Table 1: Pretraining data, models, domains, and datasets we experiment with.

| Pretraining Data | Model | Domain | Dataset |
|---|---|---|---|
| LAION2B-en | SD1.1 | Human Faces 🧑 Art Style 🖼️ | Celebrities, Politicians Classical, Modern artists |
| | SD1.5 | Human Faces 🧑 Art Style 🖼️ | Celebrities, Politicians Classical, Modern artists |
| LAION-5B | SD2.1 | Human Faces 🧑 Art Style 🖼️ | Celebrities, Politicians Classical, Modern artists |

Table 2: Prompts used to generate images of human faces (celebrities and politicians) and art styles. We generate 200 images per concept using different random seeds (1,000 images per concept). 'X' is replaced with the concept.

| # | Human faces 🧑 | Art style 🖼️ |
|---|---|---|
| 1. | A photorealistic close-up photograph of X | A painting in the style of X |
| 2. | High-resolution close-up image of X | An artwork in the style of X |
| 3. | Close-up headshot of X | A sketch in the style of X |
| 4. | X's facial close-up | A fine art piece in the style of X |
| 5. | X's face portrait | An illustration in the style of X |

**Text-to-image Models and Training Data.** We use Stable Diffusion (SD) as the text-to-image models [38].

Specifically we use SD1.1, SD1.5 that were trained on LAION2B-en, a 2.3 billion image-caption pairs dataset, filtered to contain only English captions. In addition, we use SD2.1 that was trained on LAION-5B, a 5.85 billion image-text pairs dataset, which includes LAION2B-en, and other image-caption pairs from other languages [44].

**Domains and Concepts.** We experiment with two domains – *art styles* 🖼️ and *human faces* 🧑 that are of high importance for privacy and copyright aspects of text-to-image models. Figures 1 and 2a show examples of real and generated images of art styles and human faces.

We collect a two sets of artists for the art style - classical artists and modern artists, and two sets for human faces - celebrities and politicians. Then, for each set we sample 400 names that cover a wide frequency range over the pretraining data. We provide details of the sources used to collect the concepts, and sampling procedure in Appendix M. Table 1 summarizes the pretraining data, models, domains and constructed datasets we use in this work.

**Image Generation.** We generate images for each domain by prompting models with five prompts (Table 2). We design domain-specific prompts that encourage the concepts to occupy most of the image, which simplifies the imitation measurement. We generate 200 images per concept using different random seeds for each prompt, a total of 1,000 images per concept.

## 5  Proposed Methodology: MIMETIC[2]

We illustrate our proposed methodology in Figure 3. At a high level, for a specific domain, MIMETIC[2] estimates the frequency of each concept in the pretraining data (Section 5.1) and the model's ability to imitate it (Section 5.2). We then sort the concepts based on their estimated frequencies, and find the imitation threshold using a change detection algorithm (Section 5.3).

### 5.1  Concept Frequency

**Challenges.**
Determining a concept's frequency in a multimodal dataset can be achieved by employing a high-quality classifier for that concept over every image and counting the number of detected images. However, given the scale of modern datasets with billions of images, this approach is expensive and time consuming. Instead, we make a simplifying assumption that a concept is present only if the image's caption mentions it. While this assumption does not hold in general, it is a reasonable simplification for the domains we focus on. We further discuss this assumption and provide supporting evidence for its accuracy in Appendix D. In addition, concepts often do not appear in the corresponding images, even when they are mentioned captions. For instance, Figure 2b showcases images whose captions contain "Mary Lee Pfeiffer", but her image does not always include her. On average, we find that concepts occur only in 60% of the images whose captions mention the concept.

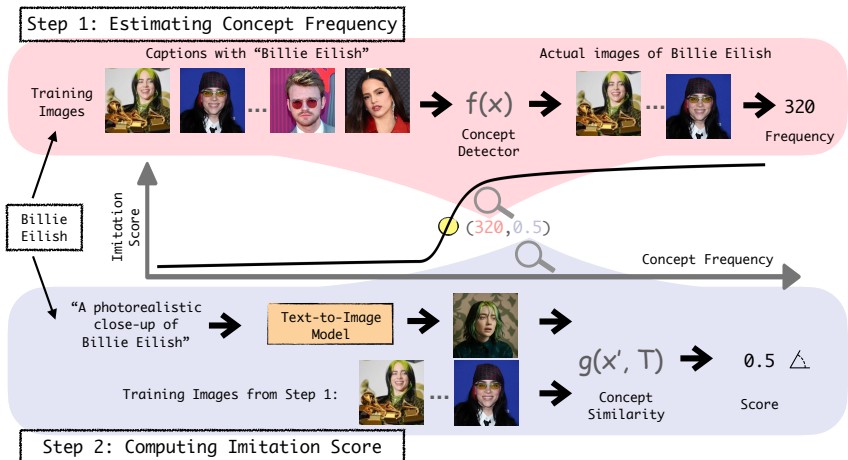

Figure 3: Overview of MIMETIC²'s methodology to estimate FIT. In Step 1, we estimate the frequency of each concept in the pretraining data by obtaining the images ($x$) that contain the concept of interest. In Step 2, we use the images of each concept and compare them to the generated images to measure imitation (using $g$ that receives reference images $T$, and generated image $x'$). We repeat this process for each concept to generate the imitation graph, and then determine the *imitation threshold* with a detection change algorithm.

**Estimating Concept Frequency** Due to the challenges described above, we start by retrieving all images whose captions mention the concept of interest and filter out the ones that do not have that concept, as detected by a classifier. We retrieve these images using WIMBD [12], a search tool based on a reverse index that efficiently finds documents (captions) from LAION containing the search query (concept). In addition, for each concept, we construct a set of high quality reference images. For example, a set of images with only the face of a single person (e.g., Brad Pitt). We collect these images automatically using a search engine, followed by a manual verification to vet the images (see Appendix E for details). Overall, we collect up to ten reference images per concept. These images are used as gold reference for automatic detection of these concepts in the images from the pretraining datasets.

Next, to classify whether a candidate image from the pretraining data contains the concept of interest, we embed the candidate image and the concept's reference images using an image encoder and measure the similarity between the embeddings. We use a face embedding model [11] for faces and an art style embedding model [51] for art style. If the similarity between a candidate image and any of the reference images is above some threshold, we consider that image to contain that concept. This threshold is established by measuring the similarity between images of the same concepts and images of different concepts which maximizes the true positive, and minimizes false positive. We provide additional details on the exact thresholds per dataset and how to find them in Appendices F and G.

Finally, we employ the classifier on all candidate images corresponding to a concept, and take those that are classified as positive. For each concept, we randomly retrieve up to 100K images whose captions mention that concept. We use the ratio of positive predictions from the retrieved candidate images and multiply it by the total caption counts of the concept in the dataset and use that as the concept frequency estimate. For concepts with less than 100K candidate images, we simply use all the images that are positively classified. Note that several URLs in the LAION datasets are dead, a common phenomenon for URL based datasets ("link rot" [7, 23]). On average, we successfully retrieved 74% of the candidate images.

## 5.2 Computing Imitation Score

**Challenges.** Computing the imitation score entails determining how similar a concept is in a generated image compared to its source images from the training data. Several approaches were proposed to accomplish this task, such as FID and CLIPScore [15, 16, 34, 39, 41]. To measure similarity, these approaches compute the similarity between the distributions of the embeddings of the generated and training images of a concept. The embeddings are obtained using image embedders like Inception model in case of FID and CLIP in case of CLIPScore. These image embedders often perform reasonably well in measuring similarity between images of common objects which constitutes most of their training data. However, they cannot reliably measure the similarity between two very similar

Table 3: *Imitation Thresholds* for human face and art style imitation for the different text-to-image models and datasets we experiment with.

| Pretraining Dataset | Model | Human Faces 👨 | | Art Style 🖼 | |
|---|---|---|---|---|---|
| | | Celebrities | Politicians | Classical Artists | Modern Artists |
| LAION2B-en | SD1.1 | 364 | 234 | 112 | 198 |
| | SD1.5 | 364 | 234 | 112 | 198 |
| LAION-5B | SD2.1 | 527 | 369 | 185 | 241 |

concepts like the faces of two individuals or art style of two artists [1, 15, 19, 51]. Therefore, MIMETIC[2] uses domain specific image embedders to measure similarity between two concepts. It uses a face embedding model [11] for measuring face similarity and an art style embedding model [51] for measuring art style similarity. Even the specific choice of these models is crucial. For instance, in early experiments we used Facenet [43], and observe it struggles to distinguish between individuals of certain demographics, causing drastic differences in the imitation scores between demographics. We provide more details on these early experiments in Appendix L, and show that our final choice of embedding models work well on different demographics.

**Estimating Imitation Score** To measure imitation we embed the generated images and training images of a concept (obtained from Section 5.1) using the concept specific image embedder. For measuring face imitation, we use InsightFace, a face embedding model [10] that extracts the individual's face from an image and generates an embedding for it. For measuring art style imitation, we use CSD, an art style embedding model [51] that generates an embedding for an image of an art work. We obtain the embeddings of the generated and training images for both the domain, and measure imitation by computing the cosine similarity between them.

To ensure that the automatic measure of similarity correlates with human perception, we also conduct experiments with human subjects and measure the correlation between the similarities obtained automatically and in the human subject experiments. We find a high correlation between the two measures of similarity (§C in Appendix).

### 5.3 Detecting the *Imitation Threshold*

After computing the concept frequencies and the imitation scores for each concept, we sort them in an ascending order of their image counts. This generates a sequence of points, each of which is a pair of image counts and imitation score of a concept. We apply a standard change detection algorithm, PELT [21], to find the image frequency where the imitation score significantly changes. Change detection is a classic statistical problem for which the objective is to find the points where the mean value of a stochastic time-series signal changes significantly. Several algorithms were proposed for change detection [55]. We choose PELT because of its linear time complexity in computing the change point. We choose the first change point as the imitation threshold (see Appendix I for details about all change points). The application of change detection assumes that increasing the image counts beyond a certain threshold leads to a large jump in the imitation scores, and we find this assumption to be accurate in our experimental results.

## 6 Results: The *Imitation Threshold*

We apply MIMETIC[2] to estimate the imitation threshold for each model-data pair, and present the results in Table 3. The imitation thresholds for SD1.1 on celebrities and politicians are 364 and 234 respectively. And the imitation thresholds for classical and modern artists are 112 and 198 respectively. Interestingly, SD1.1 and SD1.5 have the same thresholds for all the four datasets. Notably, both SD1.1 and SD1.5 are trained on LAION2B-en. The imitation thresholds for SD2.1, which is trained on the larger LAION-5B dataset is higher than the thresholds for SD1.1 and SD1.5. The imitation threshold for SD2.1 on celebrities and politicians are 527 and 369 respectively, and on classical and modern artists are 185 and 241 respectively. We hypothesize that the difference in performance of SD2.1 and SD1.1 is due to the difference in their text encoders [32]. (The difference in performance of SD2.1 and SD1.5 was also reported by several users on online forums.) To test this hypothesis, we compute the imitation thresholds for politicians for all SD models in series 1: SD1.1, SD1.2, SD1.3, SD1.4, and SD1.5. We found that the imitation thresholds for all these models are almost the same. We present the graphs for all these models in Appendix H.

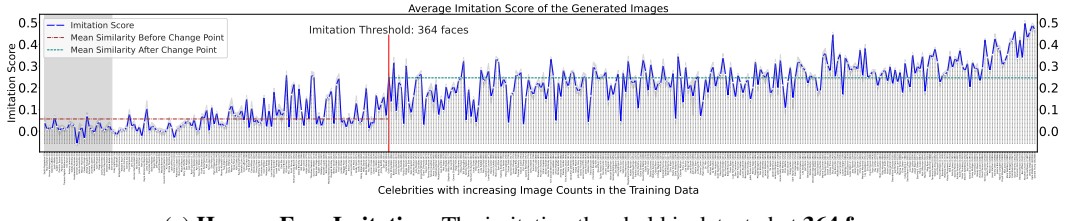

(a) **Human Face Imitation.** The imitation threshold is detected at **364 faces**.

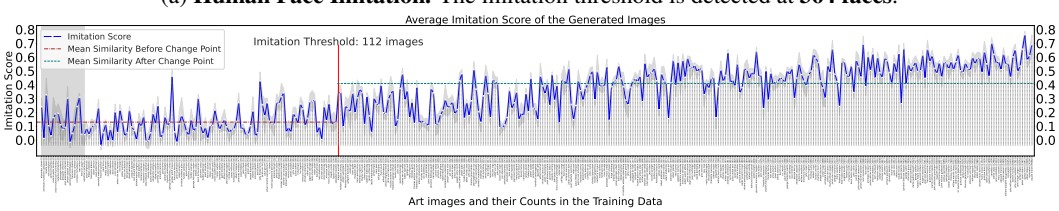

(b) **Art Style Imitation.** The imitation threshold is detected at **112 images**.

Figure 4: **Human Face** and **Art Style** imitation graphs of SD1.1 for the celebrity and classical artists datasets. The x-axis represents the sorted counts from the training set (and each concept), and the y-axis represents the similarity between the training and generated images. Concepts with zero image frequencies are shaded with light gray. We show the mean and variance over the five generation prompts. The red vertical line indicates the imitation threshold, and the horizontal green line represents the similarity threshold.

Note that celebrities have a higher imitation threshold than politicians. We hypothesize this happens due to inherent differences in the data distribution in these two datasets, which makes it harder to learn the concept of celebrities than politicians. To test this hypothesis, we compute the average number of images with a single person for people with less than 1,000 images in the pretraining dataset. We find that politicians have about twice the number of single person images compared to celebrities. As such, images that have only the concept of interest increase the ability of the model to learn from them, thus lowering the imitation threshold. We observe a similar pattern with artists: the imitation threshold for modern artists is higher than for classical artists.

We also present the plots of the imitation scores as a function of the image frequencies of the concepts in the three datasets. Figures 4a and 4b show the imitation graphs of celebrities and art styles, respectively for SD1.1. The x-axis describes the sorted concept frequency and the y-axis describes the imitation score (averaged over the five image generation prompts). We showcase the graphs for the other models and domain in Appendix J, which follow similar trends.

In Figure 4a, we observe that the imitation scores for individuals with low image frequencies are close to 0 (left side), and increase as the image frequencies move towards the right side. The highest similarity is 0.5 and it is for individuals in the rightmost region of the plot. We observe a low variance in the imitation scores across prompts. We also note that the variance does not depend on the image frequencies ($0.0003 \pm 0.0005$) – indicating that the performance of the face embedding model does not depend on the popularity of the individual.

Similarly, in Figure 4b, we observe that imitation scores for art styles with low image frequencies are close to 0.2 (left side), and increase as the image frequencies move towards the right side. The highest similarity is 0.76 and it is for the artists in the rightmost region of the plot. We also observe a low variance across the generation prompts, and the variance does not depend on the image frequency of the artist ($0.003 \pm 0.003$).

**Results Discussion.** Overall, we observe that the imitation thresholds are similar across the different image generation models and pretraining datasets, but are domain dependent. They show little variance across various image generation prompts. And most importantly, the thresholds computed by MIMETIC[2] have a high degree of agreement with human perception of imitation.

We also note the presence of several outliers in both plots, that can be categorized into two types: (1) concepts whose image counts are smaller than the imitation threshold, but their imitation scores are considerably high; and (2) concepts whose image counts are higher than the imitation threshold, but their imitation scores are low. As such, from a privacy perspective, the first kind of outliers are more crucial than the second ones This is because the imitation threshold should act as guarantors of privacy. It would be fine if a concept with a frequency higher than the threshold is not imitated by

the model (false positive), but it would be a privacy violation if a model can imitate a concept with frequency lower than the threshold (false negative). Therefore, it is preferable to underestimate the imitation threshold to minimize false negatives. Upon further analysis, we find that the actual concept frequencies of *all the false negative outliers* is much higher than what MIMETIC$^2$ counts, primarily due to aliases of names, thereby alleviating the privacy violation concerns (see Appendix B).

We also note that the range of the imitation scores of different domains have different y-axis scales. This is due to the difference in embedding models used in both cases. The face embedding model can distinguish between two faces much better than the art style model can distinguish between two styles (see Appendices F and G), and therefore the scores for the concepts on the left side of the imitation threshold is around 0 for face imitation and 0.2 for style imitation. The face embedding model also gives lower score to the faces of the same person, compared to the style embedding model's score for images of the same art styles, and therefore the highest scores for face imitation is 0.5, whereas it is 0.76 for art style imitation. However, the absolute values on the y-axis do not matter for estimating the imitation threshold as long as the trend is similar, which is the case for both domains.

# 7 Discussion and Limitations

**Equal Effect Assumption.** An assumption in the formulation of MIMETIC$^2$ is that every image of a concept contributes equally to the learning of the concept. However, not all images are created equal. While analyzing celebrities' images for instance, we often find that individuals whose images are mostly close-ups of a single person have a higher imitation score than individuals whose images are cluttered by multiple people, since concept-centered images enhance their learnability.

We hope to investigate this assumption in future work, and address this, and other potential confounders.

**Factors Affecting the Imitation Threshold.** In this work we attribute the imitation of a concept to its image count. However, image count – although a crucial factor – is not the only factor that affects imitation. Several other factors like image resolution, alignment between images and their captions, the variance between images of a concept, etc., may affect imitation.

Several training time factors like the optimization objective, learning schedule, training data order, model capacity, model architecture also affect the imitation threshold. We discuss the difference in the imitation thresholds of SD1.1, SD1.5 and SD2.1 is attributed to the difference in their text encoders. SD1.1 and SD1.5 use CLIP model [35] as their text encoder and SD2.1 uses OpenCLIP [18] as its text encoder. Note that while these may impact the behavior of the model, our work is interested in a particular model-data pair, for which we investigate. We do not claim that our results would generalize to other models, or datasets, and leave the question on how to FIT that generalize across models to future work.

# 8 Conclusions

Text-to-image models can imitate their training images [5, 49, 50]. This behavior is potentially concerning because these models' training datasets often include copyrighted and licensed images. Imitating such images would be grounds for violation of copyright and privacy laws. In this work, we seek to find the number of instances of a concept that a text-to-image model needs in order to imitate it – the *imitation threshold*. We posit this as a new problem, **F**inding the **I**mitation **T**hreshold (FIT) and propose an efficient method for finding such threshold. Our method, MIMETIC$^2$, utilizes pretrained models to estimate the imitation threshold for human face and art style imitation using three text-to-image models trained on two different pretraining datasets. We find the imitation threshold of these models to be in the range of 200-600 images depending on the setup. As such, on the domains we evaluate in this work trained on our models, our results indicate that models cannot replicate concepts that appear less than 200 times in the training data.

By estimating the imitation threshold, we provide insights on successful concepts imitation based on their training frequencies. Our results have striking implications on both the text-to-image models users and providers. These thresholds can inform text-to-image model providers what concepts are in risk of being imitated, and on the other hand, serve as a basis for copyright and privacy complaints.

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

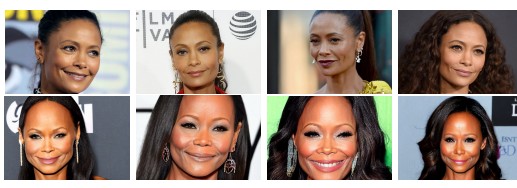 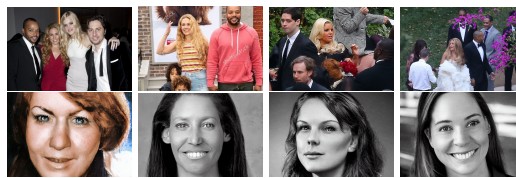

(a) **Outlier Category 1.** *Thandiwe Newton* is aliased as *Thandie Newton*, which leads to lower counts of her images in the dataset since MIMETIC[2] only collects images whose caption mentions *Thandiwe Newton*.

(b) **Outlier Category 2.** Most of the images whose captions mention *Cacee Cobb* have multiple people in them, only 6 images have her as the only person, leading to a low imitation score in generated images.

Figure 5: Examples of the two categories of outliers. The top and bottom rows show the real and SD1.1 generated images respectively. The images were generated using the following prompt: "a photorealistic close-up image of *[name].* "

## A  Additional Related Work

**Imitation in Text-to-Image Models:**    Carlini et al. [5], Somepalli et al. [49] demonstrated that diffusion models can memorize and imitate duplicate images from their training data (they use 'replication' to refer to this phenomenon). Casper et al. [8] corroborated the evidence by showing that these models imitated art styles of 70 artists with high accuracy (as classified by a CLIP model) when prompted to generated images in their styles (a group of artists also sued Stability AI claiming that their widely-used text-to-image models imitated their art style, violating copyright laws [42]). However, these works did not study how much repetition of a concept's images would lead the model to imitate them. Studying this relation is important as it serves to guide institutions training these models who want to comply with copyright and privacy laws.

**Mitigation of Imitation in Text-to-Image Models:**    Several works proposed to mitigate the negative impacts of text-to-image models. Shan et al. [48] proposed GLAZE that adds imperceptible noise to the art works such that diffusion models are unable to imitate artist styles. A similar approach was proposed to hinder learning human faces [47]. Wang et al. [57] proposed adding noise to training images, which can be used to detect if a model has been trained on those images. Lu et al. [28] propose pushing the generated images away from the distribution of training images to minimize mitigation. Gandikota et al. [13], Kumari et al. [22] proposed algorithms to remove specific styles, explicit content, and other copyrighted material learned by text-to-image models. On a related note, Xie et al. [59] proposed Diffusion-ReTrac that finds training images that most influenced a generated image, and thereby provide a fair attribution to training data contributors.

## B  Analysis: Investigating Outliers

The imitation score plots in the previous section, while showcasing a clear trend, have several outliers. In this section, we analyze the imitation scores for such outliers, where we present two examples in Figure 5 (additional outliers can be found in Figures 28 and 29 in the appendix).

**Low Image Counts and High Imitation Scores.**  Figure 5a shows an example of such a case: *Thandiwe Newton*'s image count is 172 in LAION2B-en, lower than the *imitation threshold* for celebrities: 364. However, her imitation score of 0.26 is much higher than those of neighboring celebrities with similar image counts (with scores of 0.01 and 0.04). Further investigation reveals that Thandiwe Newton is also known as *Thandie Newton*. Since this alias may also be used to describe her in captions, MIMETIC[2] may have underestimated her image counts. We repeat the process for estimating the image counts with the new alias, and find that *Thandie Newton* appears in 12,177 images, bringing the cumulative image count to 12,349, which significantly surpasses the established imitation threshold. The two aliases, whose total image count is considerably higher than the imitation threshold, differ by only a single letter and are similarly represented by the model's encoder (cosine similarity of 0.96), which explains the high imitation score. We find that most of the celebrities from the first kind of outliers are also known by other names which lead to underestimating their image counts. For example, *Belle Delphine* (394 images) also goes by *Mary Belle* (310 images, for a total of 704, and *DJ Kool Herc* (492 images) also goes by *Kool Herc* (269 images, for a total of 761).

The aliases explanation also largely explains the outliers in art style imitation. For instance, artist *Gustav Adolf Mossa* (19 images) also goes by just *Mossa* (15850 images), artist *Nicolas Toussaint*

*Charlet* (78 images) also goes by just *Nicolas Toussaint* (533 images), and artist *Wilhelm Von Kaulbach* (81 images) also goes by *Von Kaulbach* (978 images). See Figures 30 and 31 in the appendix for the real and generated images of these artists.

**High Image Counts and Low Imitation Scores** Several celebrities have higher image counts than the imitation threshold, but low imitation scores. Unlike the previous case, we were unable to find a common cause that explains all these outliers. However, we find explanations for specific cases. For example, a staggering proportion of the training images for several celebrities have multiple people in them. For example, out of the 706 total images of *Cacee Cobb*, only 6 images have her as the only person in the image (see Figure 5b). Similarly, out of 1,296 total images of *Sofia Hellqvist*, only 67 images have her as the only person and out of the 472 total images of *Charli D' Amelio*, only 82 images have him as the only person. We hypothesize that having multiple concepts in an image impedes the proper mapping of the concept's text embedding to its image embedding, which can explain the low imitation score for these concepts. We leave it to future work to further study the connection between the number of concepts in an image and models' ability to imitate these concepts.

## C    Human Perception Evaluation.

To determine if the automatic measure of similarity between the generated and training images correlate with human perception, we conduct experiments with human subjects. We asked the participants to rate generated images on the Likert scale [26] of 1-5 based on their similarity to real images of celebrities, the same ones used for measuring the imitation score. The participants were not informed of the research objective of this work.

For human face imitation, we conduct this study with 30 participants who were asked to rate 10 (randomly selected) generated images for a set of 40 celebrities. To determine the accuracy of the imitation threshold estimated by MIMETIC$^2$, we select the celebrities such that half of them have image frequencies below the threshold and the other half above it. We measure the Spearman correlation [60] between the imitation scores computed by the model and the ratings provided by the participants. Due to the variance in perception, we normalize the ratings from the participants before computing the average rating for generated images of a celebrity. The Spearman correlation between human perception and the imitation scores is **0.85**, signifying a high quality imitation estimator. We also measure the agreement between the imitation threshold that MIMETIC$^2$ estimates and the threshold that humans perceive. For this purpose, we convert the human ratings to binary values and treat it as the ground truth (any rating of 3 or more is treated as 1 and less than 3 is treated as 0). As for the MIMETIC$^2$' predictions, we construct another set of the same size that has a zero for a celebrity whose concept frequency is lower than the imitation threshold, and 1 otherwise. To measure the agreement, we compute the element-wise dot product between these two sets. We find the agreement to be 82.5%, signifying a high degree of agreement for MIMETIC$^2$'s automatically computed threshold.

For art style imitation, we conduct this study with an art expert due to the complexity of detecting art styles. The participant was asked to rate five generated images for 20 art styles, half of which were below the imitation threshold and the other half, above the threshold. We find the Spearman correlation between the two quantities to be **0.91** – demonstrating that our imitation scores are highly correlated with an artist's perception of style similarity. Similar to the previous case, we measure the agreement of the imitation threshold, which we find to be 95% – signifying a high degree of agreement for MIMETIC$^2$'s computed threshold.

## D    Caption Occurrence Assumption

For estimating the concept's counts in the pretraining dataset we make a simplifying assumption: a concept can be present in the image only if it is mentioned in a paired caption. While this assumption isn't true in general, we show that for the domains we experiment with, it mostly holds in practice.

For this purpose, we download 100K random images from LAION2B-en, and run the face detection (used in Section 5) on all images, and count the faces of the ten most popular celebrities in our sampled set of celebrities. Out of the 100K random images, about 57K contain faces. For each celebrity, we compute the similarity between all the faces in the downloaded images and the faces in the reference images of these celebrities. If the similarity is above the threshold of 0.46, we consider that face to belong to the celebrity (this threshold is determined in Appendix F to distinguish if two images are of the same person or not). Table 4 shows the number of faces we found for each celebrity

Table 4: Face count of the ten most popular celebrities in 100K random LAION images. The small percentage of the images we miss shows that our assumption of counting the images where a concept is mentioned in the caption is empirically reasonable.

| Celebrity | Face Count in 100K images | Face Count in Images with Caption Mention | Percentage of Missed Images | Number of Missed Images |
|---|---|---|---|---|
| Floyd Mayweather | 1 | 0 | 0.001% | 23K |
| Oprah Winfrey | 2 | 0 | 0.002% | 46K |
| Ronald Reagan | 6 | 3 | 0.003% | 69K |
| Ben Affleck | 0 | 0 | 0.0% | 0 |
| Anne Hathaway | 0 | 0 | 0.0% | 0 |
| Stephen King | 0 | 0 | 0.0% | 0 |
| Johnny Depp | 9 | 1 | 0.008% | 184K |
| Abraham Lincoln | 52 | 1 | 0.051% | 1.17M |
| Kate Middleton | 34 | 1 | 0.033% | 759K |
| Donald Trump | 16 | 0 | 0.016% | 368K |

in the 100K random LAION images. We also show the face counts among these images whose captions mention the celebrity. We find that 1) the highest frequency an individual appears in an image without their name mentioned in the caption is 51 (*Abraham Lincoln* is mentioned once in the caption and he appears a total of 52 times), and 2) the highest percentage of image frequency that we miss is 0.051%, and 3) most of the other miss rates are much smaller (close to 0). Such low miss rates demonstrate that our assumption of counting images when a concept is mentioned in the caption is empirically reasonable.

We also note that this assumption would fail if we were computing image frequencies for concepts that are so widely common that one would not even mention them in a caption, for example, phone, shoes, or trees.

# E  Collection of Reference Images

## E.1  Collection of Reference Images for Human Faces Domain

The goal of collecting reference images is to use them to filter the images of the pretraining dataset. These images are treated as the gold standard reference images of a person and images collected from pretraining dataset are compared to these images. If the similarity is higher than a threshold then that image is considered to belong to that person (see Section 5 for details). We describe an automatic manner of collecting the reference images. The high level idea is to collect the images from Google Search and automatically select a subset of those images that are of the same concept (same person's face or same artist's art). Since this is a crucial part of the overall algorithm, we manually vet the reference images for all the concepts to ensure that they all contain the same concept.

**Collection of Reference Images for Human Face Imitation:**  We collect reference images for celebrities and politicians using a three step process (also shown in Algorithm 1):

1. **Candidate set:**  First, we retrieved the first hundred images by searching a person's name on Google Images. We used SerpAPI [46] as a wrapper perform the searches.

2. **Selecting from the candidate set:**  Images retrieved from the internet are noisy and might not contain the person we are looking for. Therefore we filter images that contain the person from the candidate set of images. For this purpose, we use a face recognition model. We embed all the faces in the retrieved images using a face embedding model and measure the cosine similarity between each one of them. The goal is to search for a set of faces that belong to the same person and therefore will have a high cosine similarity to each other.

    One strategy is for the faces to form a graph where the vertices are the face embeddings and the edges connecting two embeddings have a weight equal to the cosine similarity between them, and we select a dense k-subgraph [24] from this graph. Selecting such a subgraph means finding a mutually homogeneous subset. We can find the vertices of this dense k-subgraph by cardinality-constrained submodular function minimization [3, 30] on a facility location function [3]. We run this minimization and select a subset of images (at least of size ten) that has the highest average cosine similarity between each pair of images.

3. **Manual verification:** Selecting the faces with the highest average similarity is not enough. This is because in many cases the largest set of faces in the candidate set are not of the person we look for, but for someone closely associated with them, in which case, the selected images are of the other person. For example, all the selected faces for *Miguel Bezos* were actually of *Jeff Bezos*. Therefore, we manually verify all the selected faces for each person. In the situation where the selected faces are wrong, we manually collect the images for them, for example, for Miguel Bezos. We collect at least 5 reference images for all celebrities.

---

**Algorithm 1:** Collection of Reference Images for Human Face Imitation

---

**Input:** Person's name P
**Output:** Verified Set of Images of P

1 $images \leftarrow$ SerpAPI(P) ; ▷ Retrieve initial image set using SerpAPI
2 $candidateSet \leftarrow$ Submodular_Minimization($images$) ; ▷ Select candidate set using submodular minimization
3 $verifiedSet \leftarrow$ manualVerification($candidateSet$) ; ▷ Manually verify the candidate set

---

**Collection of Reference Images for Art Styles** We collect reference images for each artist (each artist is assumed to have a distinct art style) from Wikiart, the online encyclopedia for art works. Since the art works of each artist were meticulously collected and vetted by the artist community, we consider all the images collected from Wikiart as the reference art images for that artist.

# F Implementation Details of MIMETIC$^2$ for Human Face Imitation

## F.1 Filtering of Training Images

Images whose captions mention the concept of interest often do not contain it (as shown with Mary-Lee Pfeiffer in Figure 2b). As such, we filter images where the concept does not appear in the image, which we detect using a dedicated classifier. In what follows we describe the filtering mechanism.

**Collecting Reference Images:**

We collect reference images for each person using SerpAPI as described in Appendix E. These images are the gold standard images that we manually vet to ensure that they contain the target person of interest (see Appendix E for the details). We use the reference images to filter out the images in the pretraining dataset that are not of this person. Concretely, for each person we use a face embedding model [10] to measure the similarity between the faces in the reference images and the faces in the images from the pretraining datasets whose captions mention this person. If the similarity of a face in the pretraining images to any of the faces in the reference images is above a certain threshold, that face is considered to belong to the person of interest. We determine this threshold to distinguish faces of the same person from faces of different persons in the next paragraph. Note that this procedure already filter outs any image that does not contain a face, because the face embedding model would only embed an image if it detects a face in that image.

**Determining Filtering Threshold:** The next step is to determine the threshold for which we consider two faces to belong to the same person. For this purpose, we measure the similarity between pairs of faces of the same person and the similarity between pairs of faces of different persons Since the reference images for each person is manually vetted to be correct, we use these images for this procedure. We plot the histogram of the average similarity between the faces of the same person (blue colored) and the similarity between faces of different persons (red colored) in Figure 6. We see that the two histograms are well separated, with the lowest similarity value between the faces of the same person being 0.56 and the highest similarity value between the faces of different persons being 0.36. Therefore any threshold value between 0.36 and 0.56 can separate two face of the same person, from the faces of different people. In our experiments, we use the midpoint threshold of 0.46 (true positive rate (tpr) of 100%; false positive rate (fpr) of 0%) to filter any face in the pretraining images that do not belong to the person of interest. The filtering process gives us both the image frequency a person in the pretraining data, and the pretraining images that we compare the faces in the generated images to measure the imitation score.

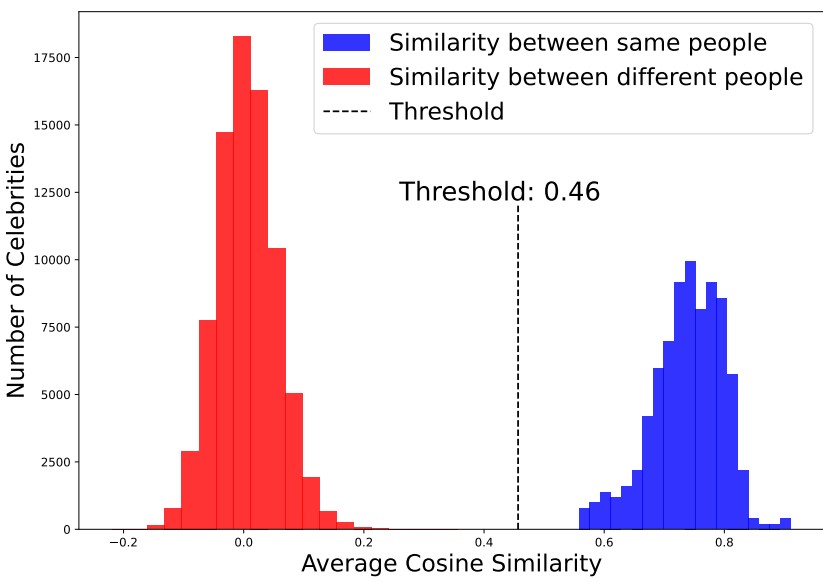

Figure 6: Average cosine similarity between the faces of the same people (blue colored) and of the faces of different people (red colored), measured across the reference images of the celebrities.

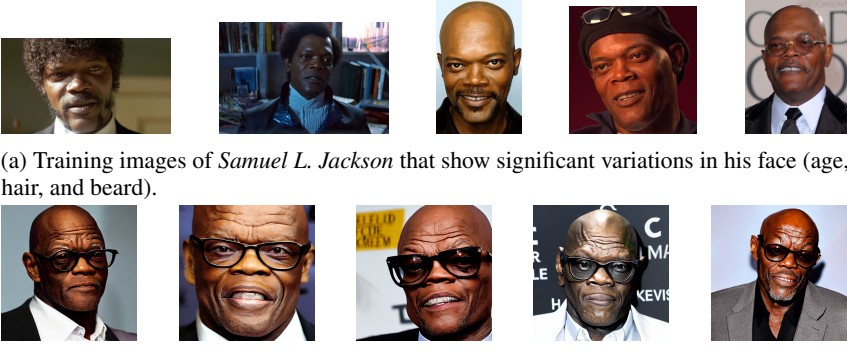

(a) Training images of *Samuel L. Jackson* that show significant variations in his face (age, hair, and beard).

(b) Generated images of *Samuel L. Jackson* that show the model has captured a specific characteristic of his face (middle-aged, bald, with no or little beard).

Figure 7: Real and generated images of *Samuel L. Jackson*.

## F.2 Measurement of Imitation Score

To measure the imitation between the training and generated images of a person, we compute the cosine similarity between the face embeddings of the faces in their generated images and their filtered training images from the previous step. However, measuring the similarity using all the pretraining images can underestimate the actual imitation. This is because several individuals have significant variations in their faces in the pretraining images and the text-to-image model does not capture all these variations. For example, consider the pretraining images of *Samuel L. Jackson* in Figure 7a. These images have significant variations in beard, hair, and age. However, when the text-to-image model is prompted to generate images of *Samuel L. Jackson*, the generated images in Figure 7b only show a specific facial characteristic of him (middle-aged, bald, with no or little beard). Since MIMETIC[2]'s goal is not to measure if a text-to-image model captures all the variations of a person, we want to reward the model even if it has only captured a particular characteristic (which it has in this case of *Samuel L. Jackson*). Therefore, instead of comparing the similarity of generated images to all the training images, we compare the similarity to only the ten training images that have the highest cosine similarity to the generated images on average.

# G  Implementation Details of MIMETIC$^2$ for Art Style Imitation

## G.1  Filtering of Training Images

For art style imitation, we consider each artist to have a unique style. We collect the images from the pretraining dataset whose captions mention the name of the artist whose art style imitation we want to measure. Similar to the case of human face imitation, we want to filter out the pretraining images of an artist that in reality was not created by that artist, but their captions mention them. We implement the filtering process in two stages. In the first stage, we filter out non-art images in the pretraining dataset (note that the captions of these images still mention the artist, but the images themselves are not art works) and in the second stage we filter out art works of other artists (the captions of these images mention the artist of interest and the image itself is also an art work, but by a different artist). The implementation details for each stage is as follows:

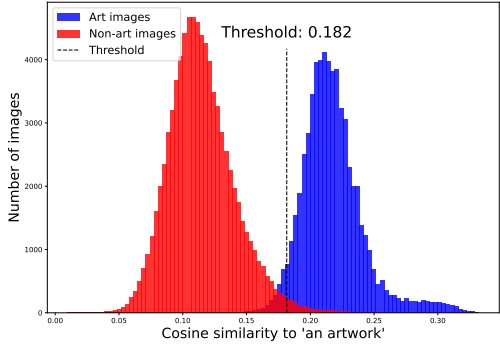
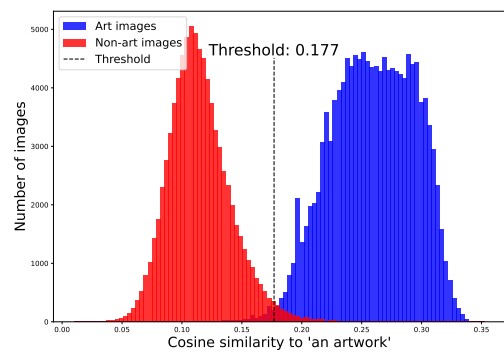

(a) Histogram of the cosine similarity of embeddings of art and non-art images to embeddings of 'an artwork' for classical artists.

(b) Histogram of the cosine similarity of embeddings of art and non-art images to embeddings of 'an artwork' for modern artists.

Figure 8: The first filtering step involves determining the threshold to distinguish between art and non-art images from the pretraining images, for which we compare the similarity of the image's embedding to the embedding of the text "an artwork".

**Filtering Non-Art Images:**  To filter non-art images from the pretraining dataset, we use a classifier that separates art images from non-art images. Concretely, we embed the pretraining images using a CLIP ViT-H/14 [18] image encoder and measure the cosine similarity of the image embeddings and the text embeddings of the string '*an artwork*', embedded using the text encoder of the same model. Only when the similarity between the embeddings is higher than a threshold described below, we consider those pretraining images as an artwork. To determine this threshold, we choose a similarity score that separates art images from non-art images. We use the images from the Wikiarts dataset [40] as the (positive) art images and MS COCO dataset images [27] as the (negative) non-art images. Note that MS COCO dataset was collected by photographing everyday objects that art was not part of, making it a valid set of negative examples of art.

We plot the histogram of cosine similarity of the embeddings of art and non-art images to the text embedding of '*an artwork*' (see Figures 8a and 8b. We observe that the art and non-art images both the artist groups are well separated (although not perfect, Figure 9 and Figure 10 shows examples of misclassified and correctly classfied images from both datasets). We choose the threshold that maximizes the F1 score of the separation (0.182 for the classical artists and 0.177 for the modern artists).

**Filtering Images of Other Art Styles:**  Similar to the case of human faces, not all art images whose captions mention an artist were created by that artist. We want to filter out such images. For this purpose, we collect reference images for each artist (see Appendix E for details) and use them to classify the training images that belong to the artist of interest. Concretely, we measure the similarity between the pretraining images and the reference images of each artist, and only retain images whose similarity to the reference images is higher than a threshold.

To determine this threshold, we measure the similarity between pairs of art images of the same artists and pairs of art images from different artists. We embed the images using an art style embedding model [51] and plot the histogram of similarities between art images of the same artist (blue colored)

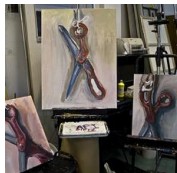 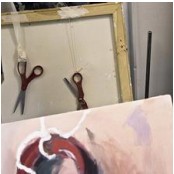 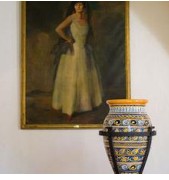 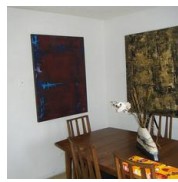 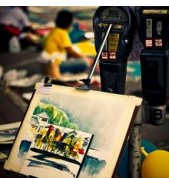

(a) Images from the MS COCO dataset that were classified as art by the threshold we choose. These images clearly have paintings in them and therefore are classified in that category. These images were selected in MS COCO for different categories like scissors, chair, parking meter, and vase.

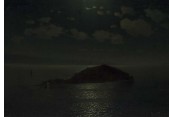 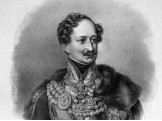 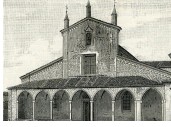 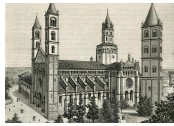 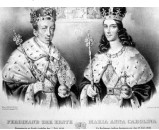

(b) Images from the Wikiarts dataset that were classified as non-art by the threshold we choose.

Figure 9: Images that are misclassified by our art vs. non-art threshold in Figure 8a.

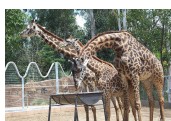 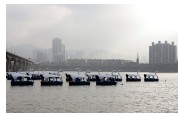 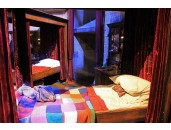 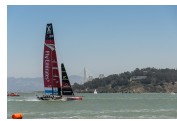 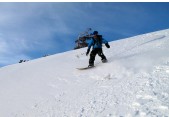

(a) Images from the MS COCO dataset that were correctly classified as non-art.

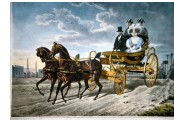 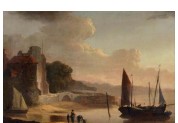 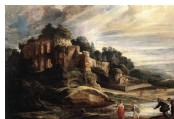 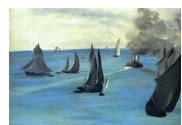 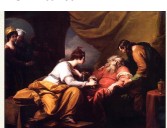

(b) Images from the Wikiarts dataset that were correctly classified as art.

Figure 10: Images that are correctly classified by our art vs. non-art threshold in Figure 8a.

and art images of different artists (red colored) in Figure 11a for classical artists and Figure 11b for modern artists. We see that the two histograms are well separated (although not perfect, Figure 12 shows paintings by two artists whose art style is very similar and cannot be distinguished by our threshold). We choose the threshold that maximizes the F1 score of the separation between these two groups (0.278 for classical artists and 0.288 for modern artists). The retained images give us both the image counts of each artist and the training images that we compare to the generated images to measure the imitation score.

### G.2 Similarity Measurement

We embed all the generated images and the filtered pretraining images using the art style embedding model [51] and measure the cosine similarity between each pair of generated and pretraining images. Similar to the case of the human faces, we do not want to underestimate the art style similarity between the generated and training images by comparing the generated images to all the training images of this artist. Therefore, we measure the similarity of generated images to the ten training images that are on average the most similar to the generated images.

## H   Imitation Thresholds of SD models in Series 1 and 2

Our experimental results in Section 6 found that for most domains the imitation thresholds for SD1.1 and SD1.5 are almost the same, while being higher for SD2.1. We hypothesized that the difference is due to their different text encoders. All models in SD1 series use the same text encoder from CLIP, whereas SD2.1 uses the text encoder from OpenCLIP. To test the validity of this hypothesis, we repeated the experiments for all models in SD1 series for politicians and computed their imitation thresholds. Table 5 shows the thresholds for the politicians. We find that the imitation thresholds for all the models in SD1 series is almost the same, and is lower than the threshold for SD2.1 model. This evidence supports our hypothesis of the difference in the text-encoders being the main reason for the difference in the imitation thresholds.

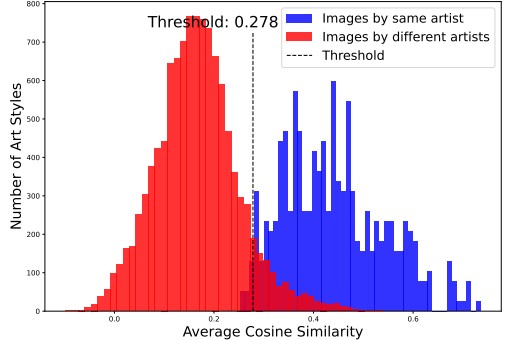 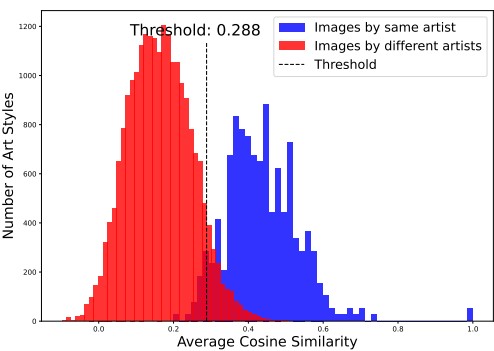

(a) Histogram of the average cosine similarity between embeddings of the images of the same artist (blue) and the art of different artists (red) for classical artists

(b) Histogram of the average cosine similarity between embeddings of the images of the same artist (blue) and the art of different artists (red) for modern artists

Figure 11: The second filtering step involves determining the if an art work whose caption mentions an artist actually belongs to that artist or not.

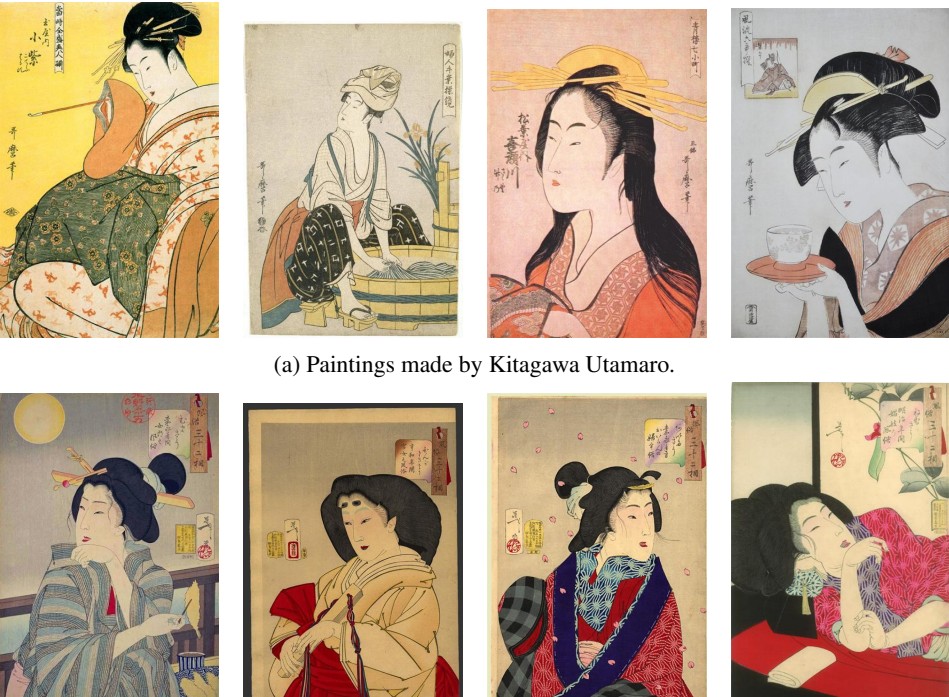

(a) Paintings made by Kitagawa Utamaro.

(b) Paintings made by Tsukioka Yoshitoshi

Figure 12: Paintings made by Kitagawa Utamaro and Tsukioka Yoshitoshi are very similar and our threshold is unable to distinguish between their styles.

## I Change Points

Table 6 we show all the change points that PELT found for each experiment (Table 3 reports the first change point as the imitation threshold).

## J All Results: The *Imitation Threshold*

In this section, we estimate the imitation threshold for human face and art style imitation for three different text-to-image models. Figure 16, Figure 17, and Figure 18 show the image counts of celebrities on the x-axis (sorted in increasing order of image counts) and the imitation score of their generated images (averaged over the five image generation prompts) on the y-axis. The images

Table 5: *Imitation Thresholds* for politicians for all models in SD1 series and SD2.1

| Pretraining Dataset | Model | Human Faces 🧑: Politicians |
|---|---|---|
| LAION2B-en | SD1.1 | 234 |
| | SD1.2 | 252 |
| | SD1.3 | 234 |
| | SD1.4 | 234 |
| | SD1.5 | 234 |
| LAION-5B | SD2.1 | 369 |

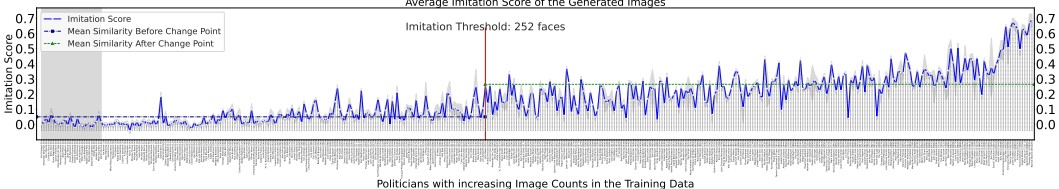

Figure 13: **Human Face Imitation (Politicians):** Similarity between the training and generated images for all politicians. The politicians with zero image counts are shaded with light gray. We show the mean and variance over the five generation prompts. The images were generated using **SD1.2**. The change point for human face imitation for politicians when generating images using SD1.1 is detected at **252 faces**.

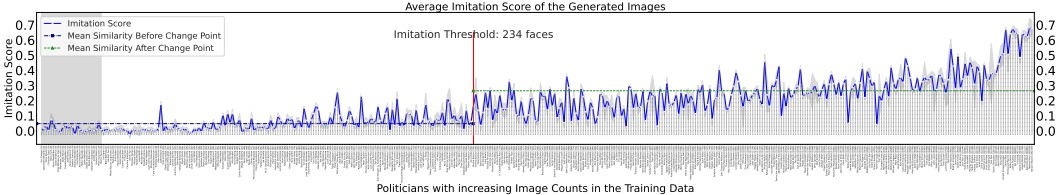

Figure 14: **Human Face Imitation (Politicians):** Similarity between the training and generated images for all politicians. The politicians with zero image counts are shaded with light gray. We show the mean and variance over the five generation prompts. The images were generated using **SD1.3**. The change point for human face imitation for politicians when generating images using SD1.1 is detected at **234 faces**.

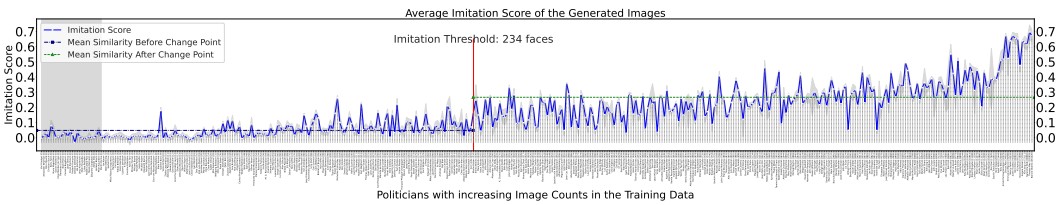

Figure 15: **Human Face Imitation (Politicians):** Similarity between the training and generated images for all politicians. The politicians with zero image counts are shaded with light gray. We show the mean and variance over the five generation prompts. The images were generated using **SD1.4**. The change point for human face imitation for politicians when generating images using SD1.1 is detected at **234 faces**.

were generated using SD1.1, SD1.5, and SD2.1 respectively. Similarity, Figure 19, Figure 20, and Figure 21 shows the image counts of the politicians and the imitation score of their generated images, for SD1.1, SD1.5, and SD2.1 respectively.

Figure 22, Figure 23, Figure 24 show the image counts of classical artists and the similarity between their training and generated images; and Figure 25, Figure 26, Figure 27 show the image counts of modern artists and the similarity between their training and generated images. The images were generated using SD1.1, SD1.5, and SD2.1 respectively.

Table 6: *Imitation Thresholds* for human face and art style imitation for the different text-to-image models and datasets we experiment with.

| Pretraining Dataset | Model | Human Faces 🧑 | | Art Style 🖼️ | |
|---|---|---|---|---|---|
| | | Celebrities | Politicians | Classical Artists | Modern Artists |
| LAION2B-en | SD1.1 | 364 | 234 | 112, 391 | 198 |
| | SD1.5 | 364, 8571 | 234, 4688 | 112, 360 | 198, 4821 |
| LAION-5B | SD2.1 | 527, 9650 | 369, 8666 | 185, 848 | 241, 1132 |

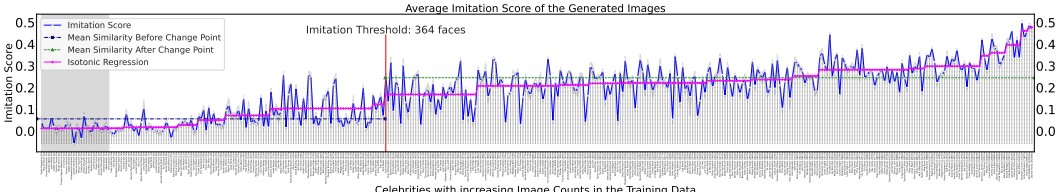

Figure 16: **Human Face Imitation (Celebrities):** Similarity between the training and generated images for all celebrities. The celebrities with zero image counts are shaded with light gray. We show the mean and variance over the five generation prompts. The images were generated using **SD1.1**. The change point for human face imitation for celebrities when generating images using SD1.1 is detected at **364 faces**.

**Imitation Threshold Estimation for Human Face Imitation:** In Figure 16, we observe that the imitation scores for the individuals with small image counts is close to 0 (left side), and it increases as the number of their image counts increase towards the right. The highest similarity is 0.5 and it is for the individuals in the rightmost region of the plot. The solid line in the plot shows the mean similarity over the five image generation prompt with the shaded area showing the variance over them. We observe a low variance in the imitation score among the generation prompts. And we also observe that the variance does not depend on the image counts which indicates that the performance of the face recognition model does not depend on the popularity of the individual. The change detection algorithm finds the change point to be at **364** faces for human face imitation for celebrities, when using **SD1.1** for image generation. Figure 17 shows the similarity between the training and generated images when images are generated using **SD1.5**. Identically to SD1.1, the change is detected at **364** faces for face imitation when using SD1.5. We also performed ablation experiments with different face embeddings models and justify the choice of our model (see Appendix L). For all the plots, we also analyze the trend by using isotonic regression which learns non-decreasing linear regression weights that fits the data best.

**Imitation Threshold Estimation for Human Face Imitation (Politicians):** Figure 19 shows the imitation scores for politicians which is very similar to the plot obtained for celebrities. We observe a low variance in the imitation score among the generation prompts. We also observe that the variance does not depend on the image counts which indicates that the performance of the face recognition model does not depend on the popularity of the individual. The change detection algorithm finds the change point to be at **234** faces for human face imitation for politicians, when using **SD1.1** for image generation. Figure 20 shows the similarity between the training and generated images when images are generated using **SD1.5**. Similar to SD1.1, the change is detected at **234** faces.

**Imitation Threshold Estimation for Art Style Imitation:** In Figure 22, we observe that the imitation scores for artists with low image counts have a baseline value around 0.2 (left side), and it increases as the number of their image counts increase towards the right. The highest similarity is 0.76 and it is for the artists in the rightmost region of the plot. We also observe a low variance across the generation prompts, and the variance does not depend on the image frequency of the artist. The change detection algorithm finds the change point to be at **112** images for art style imitation of classical artists, when using **SD1.1** for image generation. Figure 23 shows the similarity between the training and generated images when images are generated using **SD1.5**. Similar to SD1.1, the change is detected at **112** faces for art style imitation when using SD1.5. These thresholds are slightly higher for style imitation of modern artists, 198 for both SD1.1 and SD1.5.

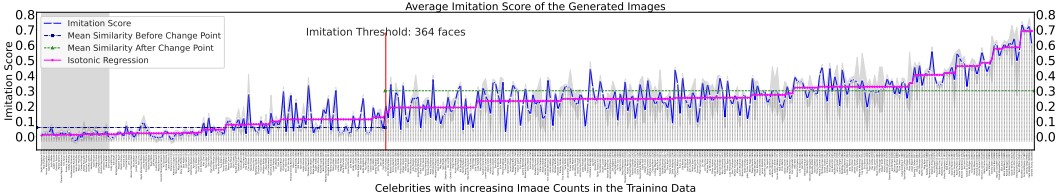

Figure 17: **Human Face Imitation (Celebrities):** similarity between the training and generated images for all celebrities. We show the mean and variance over the five generation prompts. The images were generated using **SD1.5**. The change point for human face imitation for celebrities when generating images using SD1.5 is detected at **364 faces**.

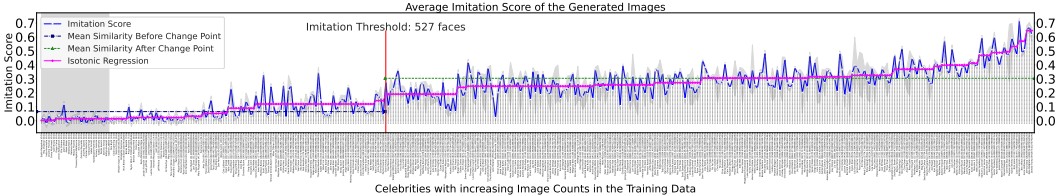

Figure 18: **Human Face Imitation (Celebrities):** similarity between the training and generated images for all celebrities. We show the mean and variance over the five generation prompts. The images were generated using **SD2.1**. The change point for human face imitation for celebrities when generating images using SD2.1 is detected at **527 faces**.

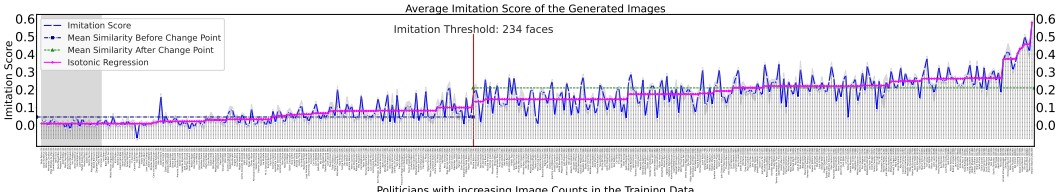

Figure 19: **Human Face Imitation (Politicians):** Similarity between the training and generated images for all politicians. The politicians with zero image counts are shaded with light gray. We show the mean and variance over the five generation prompts. The images were generated using **SD1.1**. The change point for human face imitation for politicians when generating images using SD1.1 is detected at **234 faces**.

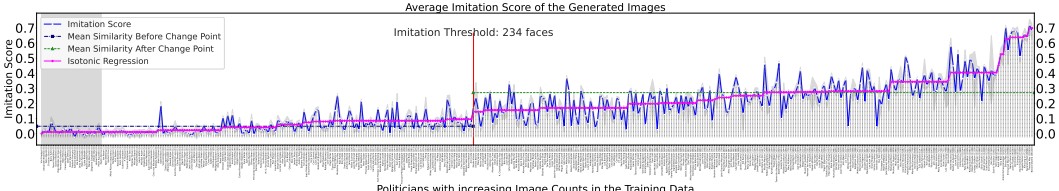

Figure 20: **Human Face Imitation (Politicians):** similarity between the training and generated images for all politicians. We show the mean and variance over the five generation prompts. The images were generated using **SD1.5**. The change point for human face imitation for politicians when generating images using SD1.5 is detected at **234 faces**.

## K   Examples of Outliers

Figure 28 and Figure 29 show examples of outliers of the first kind, where aliases of a celebrity leads to under counting of their images in the pretraining data.

Figure 30 and Figure 31 show examples of outliers of the first kind for artists, where aliases of an artist leads to under counting of their art works in the pretraining data.

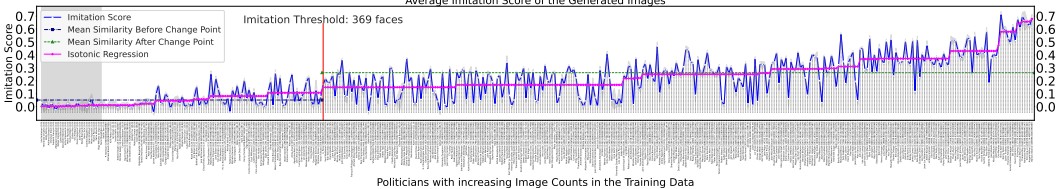

Figure 21: **Human Face Imitation (Politicians):** similarity between the training and generated images for all politicians. We show the mean and variance over the five generation prompts. The images were generated using **SD2.1**. The change point for human face imitation for celebrities when generating images using SD2.1 is detected at **369 faces**.

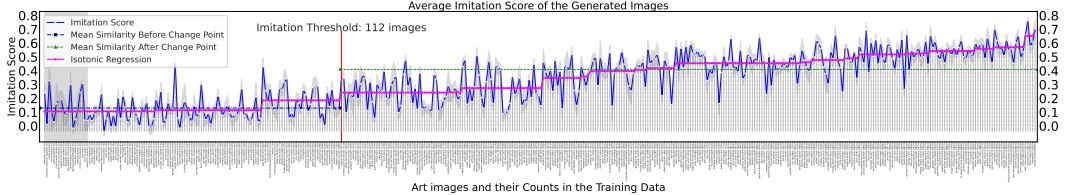

Figure 22: **Art Style Imitation (Classical Artists):** similarity between the training and generated images for **classical** art styles. We show the mean and variance over the five generation prompts. The images were generated using **SD1.1**. The change point for art style imitation when generating images using SD1.1 is detected at **112 images**.

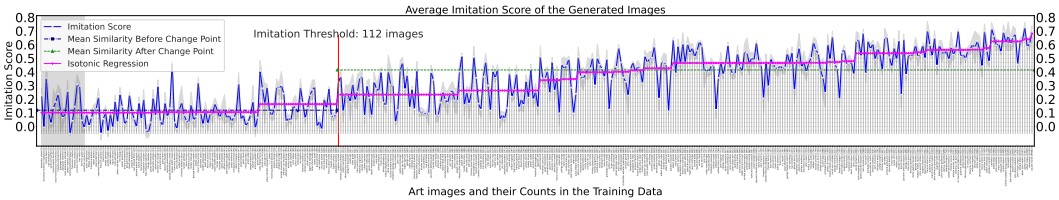

Figure 23: **Art Style Imitation (Classical Artists):** similarity between the training and generated images for **classical** art styles. We show the mean and variance over the five generation prompts. The images were generated using **SD1.5**. The change point for art style imitation when generating images using SD1.5 is detected at **112 images**.

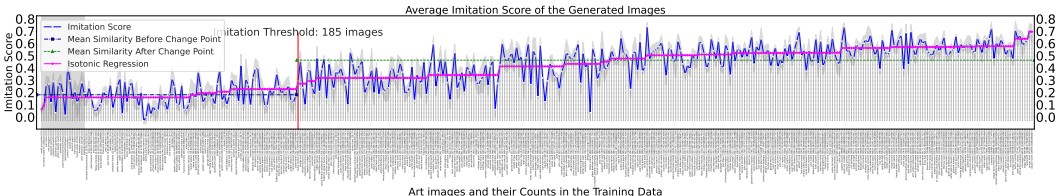

Figure 24: **Art Style Imitation (Classical Artists):** similarity between the training and generated images for **classical** art styles. We show the mean and variance over the five generation prompts. The images were generated using **SD2.1**. The change point for art style imitation when generating images using SD2.1 is detected at **185 images**.

## L  Ablation Experiment with Different Face Embedding Models

In this section, we show the difference in the performance of several face embedding models and justify the choice of the final choice of our face embedding model. Face embedding models are evaluated using two main metrics: false-match rate (FMR) and true-match rate (TMR) [31]. FMR measures how many times does a model says two people are the same when they are not and TMR measures how many times a model says two people are the same when they are the same. Ideally, a face embedding model should have low FMR and high TMR. An important variant of these metrics is the disparity of FMR and TMR of a model across different demographic groups. Ideally, a model

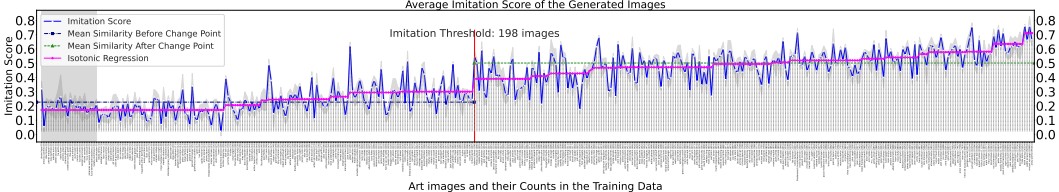

Figure 25: **Art Style Imitation (Modern Artists):** similarity between the training and generated images for **modern** art styles. We show the mean and variance over the five generation prompts. The images were generated using **SD1.1**. The change point for art style imitation when generating images using SD1.1 is detected at **198 images**.

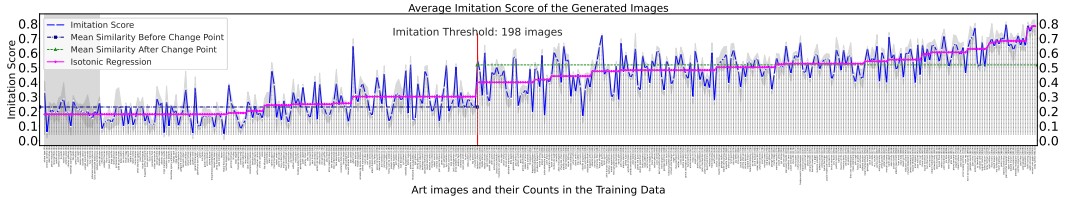

Figure 26: **Art Style Imitation (Modern Artists):** similarity between the training and generated images for **modern** art styles. We show the mean and variance over the five generation prompts. The images were generated using **SD1.5**. The change point for art style imitation when generating images using SD1.5 is detected at **198 images**.

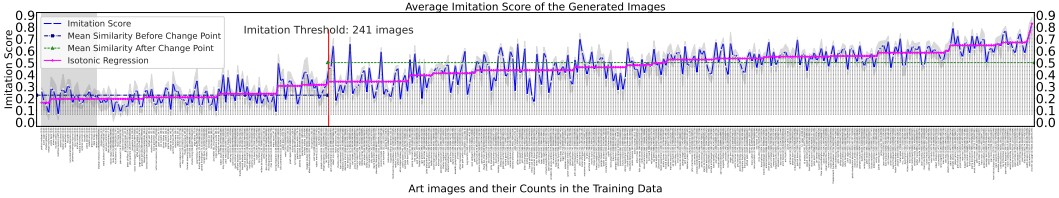

Figure 27: **Art Style Imitation (Modern Artists):** similarity between the training and generated images for **modern** art styles. We show the mean and variance over the five generation prompts. The images were generated using **SD2.1**. The change point for art style imitation when generating images using SD2.1 is detected at **241 images**.

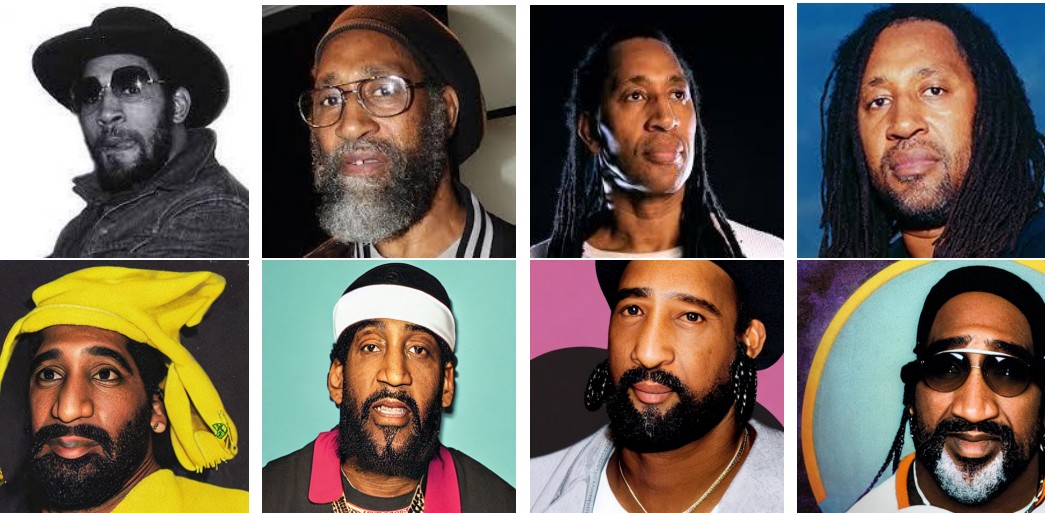

Figure 28: **Outlier Category 1: DJ Kool Herc.** *Clive Campbell* is aliased as *DJ Kool Herc*, which leads to lower counts of his images in the dataset since MIMETIC[2] only collects images whose caption mentions *DJ Kool Herc*.

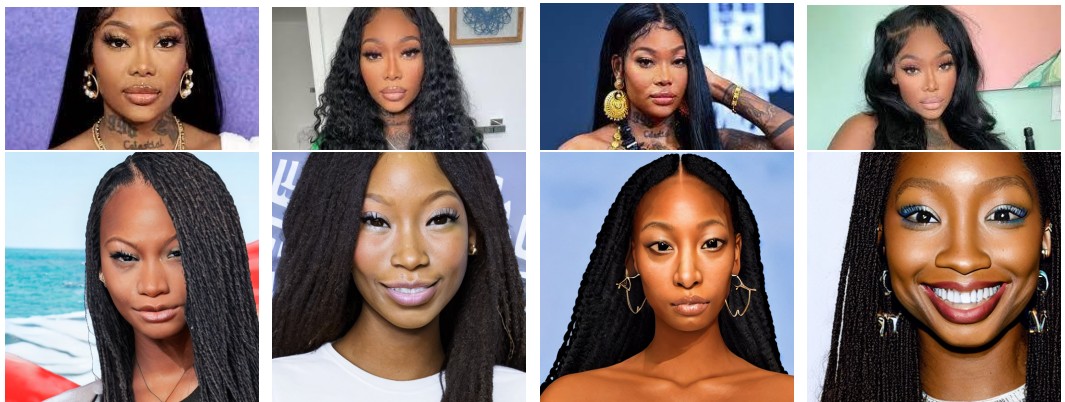

Figure 29: **Outlier Category 1: Summer Walker.** *Summer Marjani Walker* is aliased as *Summer Walker*, which leads to lower counts of her images in the dataset since MIMETIC$^2$ only collects images whose caption mentions *Summer Walker*.

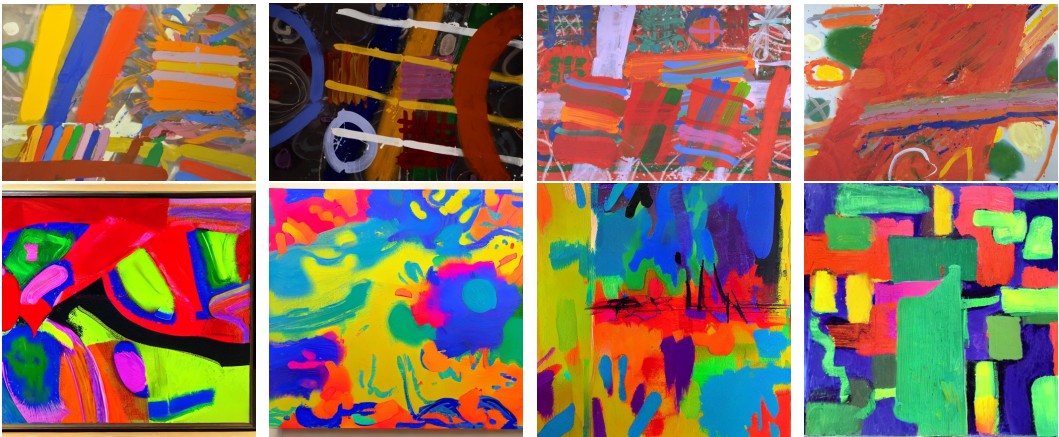

Figure 30: **Outlier Category 1: Albert Irwin.** *Albert Henry Thomas Irvin* is aliased as *Albert Irwin*, which leads to lower counts of his art images in the dataset since MIMETIC$^2$ only collects images whose caption mentions *Albert Irwin*.

should have low disparity in these metrics across different demographics. We also focus on the variance of these metrics across demographics in making the final choice.

We evaluate the FMR and TMR of eight different face embedding models (seven open-sourced and one proprietary). The open-source models were chosen based on their popularity on Github [10, 11, 45], and we also experiment with Amazon Rekognition, a proprietary model. For evaluating the disparity of these metrics across different demographic groups we grouped celebrities in six demographic groups primarily categorized according to skin color tone (black, brown, and white) and perceived gender (male and female; for simplicity). Each of the six groups had 10 celebrities (a total of 60), with no intersection between them. The categorization was done manually by looking at the reference images of the celebrities. For each celebrity, we collect 10 reference images from the internet by using the procedure described in Appendix E. We use these images to compare the FMR and TMR of the face recognition models, as these images are the gold standard images of a person.

**FMR Computation:** We compute the mean cosine similarity between the face embeddings of one individual and the faces of all other individuals in that group, and repeat the procedure for all individuals in a demographic group.

**TMR Computation:** We compute the mean cosine similarity between the embeddings of all the faces of an individuals and repeat the procedure for all the individuals in a demographic group.

Figure 32 and Figure 33 shows the FMR and TMR for six demographic groups for all the face embedding models. All the open-sourced models, except InsightFace, either have a high disparity in FMR values across the demographic groups (ArcFace, Facenet, Facenet512, DeepFace) or have

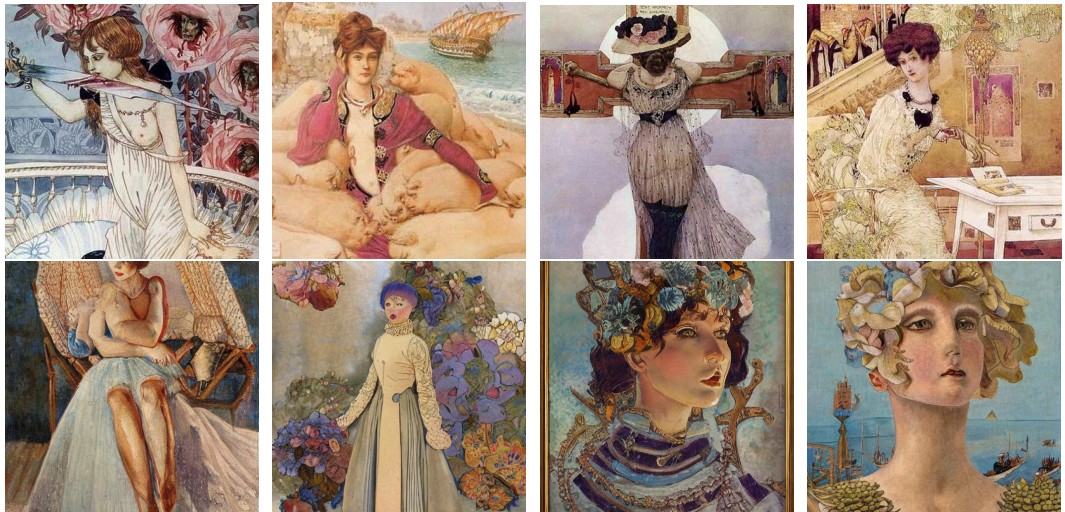

Figure 31: **Outlier Category 1: Gustav Adolf Mossa.** *Gustav Adolf Mossa* is aliased just as *Mossa*, which leads to lower counts of his art images in the dataset since MIMETIC$^2$ only collects images whose caption mentions *Gustav Adolf Mossa*.

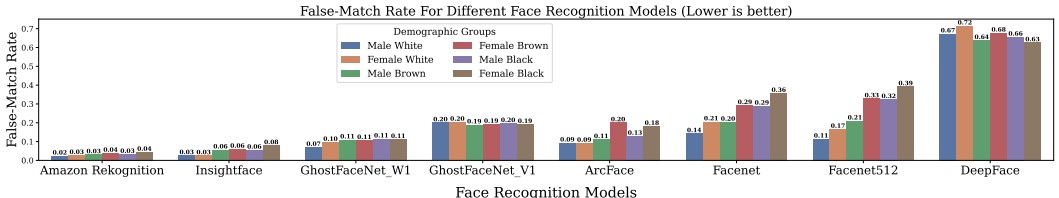

Figure 32: False-match rate (FMR) of all the face embedding models across the six demographic groups. Amazon Rekognition and InsightFace have the lowest FMR values. Moreover, these two models have lowest disparity of FMR over the demographic groups.

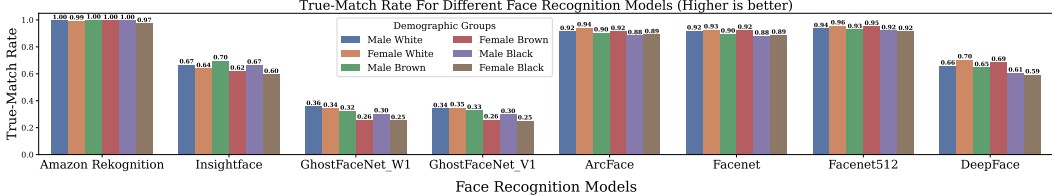

Figure 33: True-match rate (TMR) of all the face embedding models across the six demographic groups. Amazon Rekognition model has the highest TMR values.

very low TMR (GhostFaceNet_W1, GhostFaceNet_V1). We choose InsightFace for our experiments because of it has 1) a low overall FMR, 2) decent TMR, 3) a low disparity of FMR and TMR across the demographic groups, and 4) is open-sourced. Having a low disparity of the metrics across individuals of different demographic groups is crucial for an accurate estimation of the imitation threshold. The Amazon Rekognition model would also be a viable choice based on these metrics, however, it is not open-sourced and therefore expensive for our experiments.

# M  Count Distribution and the List of Sampled Entities for Each Domain

## M.1  Celebrities

We collect celebrities from `https://www.popsugar.com/Celebrities` and `https://celebanswers.com/celebrity-list/`. The distribution of the caption counts of the sampled celebrities is displayed in Table 7. The sampled celebrities in the descending order of their number of caption counts are:

Table 7: Distribution of caption counts for sampled entities in celebrities, politicians, and art styles domains.

| Caption Counts (LAION-2B) | Celebrities | Politicians | Classical Artists | Modern Artists |
|---|---|---|---|---|
| 0 | 19 | 15 | 14 | 15 |
| 1-100 | 48 | 60 | 67 | 69 |
| 100-500 | 57 | 120 | 133 | 139 |
| 500-1K | 52 | 80 | 62 | 62 |
| 1K-5K | 151 | 65 | 63 | 64 |
| 5K-10K | 19 | 40 | 39 | 32 |
| > 10K | 53 | 40 | 40 | 34 |

Donald Trump, Kate Middleton, Abraham Lincoln, Johnny Depp, Stephen King, Anne Hathaway, Ben Affleck, Ronald Reagan, Oprah Winfrey, Floyd Mayweather, Dwayne Johnson, Cameron Diaz, Cate Blanchett, Mark Wahlberg, Naomi Campbell, Nick Jonas, Jessica Biel, Kendrick Lamar, Malcolm X, Steven Spielberg, Bella Thorne, Bob Ross, Jay Leno, David Tennant, Samuel L. Jackson, Jason Statham, Mandy Moore, Victoria Justice, Scott Disick, Martin Scorsese, Ashley Olsen, Carey Mulligan, Greta Thunberg, Ashlee Simpson, Kacey Musgraves, Kurt Russell, Felicity Jones, Saoirse Ronan, Sarah Paulson, Matthew Perry, Forest Whitaker, Brendon Urie, Meg Ryan, Olivia Culpo, Joe Rogan, Sacha Baron Cohen, Terrence Howard, Natalie Dormer, Ansel Elgort, Nick Offerman, Clive Owen, Rose Leslie, Sterling K. Brown, Cuba Gooding Jr., Kevin James, Marisa Tomei, Troye Sivan, Zachary Levi, Gwendoline Christie, Hunter Hayes, Melanie Martinez, Joel McHale, Ross Lynch, Brody Jenner, Riley Keough, Robert Kraft, Ray Liotta, Eric Bana, Mark Consuelos, Chris Farley, James Garner, Lauren Daigle, Lily Donaldson, Penélope Cruz, Karen Elson, Joey Fatone, Leslie Odom Jr., Jay Baruchel, Selita Ebanks, Lana Condor, Mackenzie Foy, Doja Cat, Skai Jackson, Sofia Hellqvist, Bernard Arnault, Josh Peck, Lindsay Price, Phoebe Bridgers, Sarah Chalke, Alexander Skarsgård, Tai Lopez, Léa Seydoux, Cam Gigandet, David Dobrik, Jacob Elordi, Omar Epps, Marsai Martin, Alyson Stoner, Dree Hemingway, Gregg Sulkin, Mamie Gummer, Allison Holker, Chris Watts, Jacob Sartorius, Christine Quinn, Torrey Devitto, Alek Wek, Sandra Cisneros, Robert Irvine, Danielle Fishel, Normani Kordei, Sam Taylor Johnson, Jessica Seinfeld, Rachelle Lefevre, Joyner Lucas, Jimmy Buffet, John Wayne Gacy, Marvin Sapp, Ryan Guzman, Lindsay Ellingson, John Corbett, Michaela Coel, Hanne Gaby Odiele, Christiano Ronaldo, Scott Speedman, Addison Rae, Justice Smith, Stella Tennant, Lindsay Wagner, AJ Michalka, Charles Melton, Patricia Field, Dan Bilzerian, Annie Murphy, Michiel Huisman, Sara Foster, Diego Boneta, Danny Thomas, Oliver Hudson, Lauren Bushnell, Chris Klein, Rodrigo Santoro, Luke Hemsworth, Rhea Perlman, Michael Peña, Jodie Turner-Smith, Trevor Jackson, Jenna Marbles, Bob Morley, Zak Bagans, Liza Koshy, Steve Lacy, Nico Tortorella, Emma Corrin, Lo Bosworth, Quvenzhané Wallis, Martin Starr, David Muir, Beanie Feldstein, Lori Harvey, Eddie McGuire, Todd Chrisley, Dan Crenshaw, Amanda Gorman, Crystal Renn, Mark Richt, Magdalena Frackowiak, Danielle Jonas, Liu Yifei, Sasha Pivovarova, Ashleigh Murray, Peter Hermann, Daria Strokous, Eddie Hall, Hunter Parrish, Matt McGorry, Diane Guerrero, Simu Liu, Brady Quinn, Jill Wagner, Richard Rawlings, Sophia Lillis, Genesis Rodriguez, Diane Ladd, Frankie Grande, Olivia Rodrigo, Anwar Hadid, Hannah Bronfman, Deana Carter, Tao Okamoto, Fei Fei Sun, Taylor Tomasi Hill, Jared Followill, Margherita Missoni, Elisa Sednaoui, Thomas Doherty, Bill Skarsgård, Indya Moore, Ziyi Zhang, Cacee Cobb, Jay Ellis, Arthur Blank, Chris McCandless, Paz de la Huerta, Jacquelyn Jablonski, Michael Buffer, Annie LeBlanc, Kieran Culkin, Lacey Evans, Rachel Antonoff, Presley Gerber, Lauren Bush Lauren, Peter Firth, Tina Knowles-Lawson, Sunisa Lee, Douglas Brinkley, Hero Fiennes-Tiffin, Erin Foster, Justina Machado, Mariacarla Boscono, Summer Walker, Emma Chamberlain, Lew Alcindor, Jenna Ortega, Phoebe Dynevor, Kim Zolciak-Biermann, Allison Stokke, Malgosia Bela, Isabel Toledo, Sydney Sweeney, Mat Fraser, Hunter McGrady, Ethan Suplee, Tammy Hembrow, Ivan Moody, Danneel Harris, Marcus Lemonis, Hunter Schafer, Luka Sabbat, Sam Elliot, Kendra Spears, Stephen tWitch Boss, Joe Lacob, Tommy Dorfman, Emma Barton, Elliot Page, Sha'Carri Richardson, Barry Weiss, Julie Chrisley, Devon Sawa, Miles Heizer, Julia Stegner, Austin Abrams, Jacquetta Wheeler, Melanie Iglesias, Anna Cleveland, Eiza González, Grant Achatz, Matt Stonie, Connor Cruise, Nicholas Braun, Dan Lok, Charli D'Amelio, Jeremy Bamber, Jim Walton, Matthew Bomer, Nicola Coughlan, Una Stubbs, Andrew East, Miles O'Brien, Mary Fitzgerald, Taylor Mills, Portia Freeman, Kate Chastain, David Brinkley, Bregje Heinen, DJ Kool Herc, Barbie Ferreira, Paul Mescal, Forrest Fenn, Jamie Bochert, Yung Gravy, Daisy Edgar-Jones, Dixie D'Amelio, Jordan Chiles, Bob Keeshan, Alexandra Cooper, Kyla Weber, Chase Stokes, Belle Delphine, Joanna Hillman, Olivia O'Brien, Jillie Mack, Maggie Rizer, Sasha Calle, Tony Lopez, Danny Koker, Irwin Winkler, M.C. Hammer, Zack Bia, Alexa Demie, Bailey Sarian, Yael Cohen, Angie Varona, Trevor Wallace, Madelyn Cline, Fred Stoller, Frank Sheeran, Albert Lin, Sessilee Lopez, Zaya Wade, Maitreyi Ramakrishnan, Madison Bailey, Will Reeve, Nick Bolton, Rege-Jean Page, Matthew Garber, Yamiche Alcindor, Isaak Presley, Thandiwe Newton, Nicole Fosse, Shenae Grimes-Beech, Alex Choi, Scott Yancey, Ciara Wilson, Lexi Underwood, Manny Khoshbin, Ella Emhoff, Cole LaBrant, Wayne Carini, Greg Fishel, Ryan Upchurch, Marcus Freeman, Danielle Cohn, Sue Aikens, Kyle Cooke, Teddy Portnoy, Avani Gregg, Dan Peña, Quinton Reynolds, Eric Porterfield, Ayo Edebiri, Tara Lynn Wilson, Florence Hunt, Nicola Porcella, Pashmina Roshan, Josh Seiter, Ben Mallah, Miguel Bezos, Lukita Maxwell, Ali Skovbye, Jordan Firstman, Jeff Molina, Mary Lee Pfeiffer, Cody Lightning, Leah Jeffries, Elle Graham, Hannah Margaret Selleck, Woody Norman, Tom Blyth, Banks Repeta, Wisdom Kaye, Kris Tyson, Joey Klaasen, Tioreore Ngatai-Melbourne, Jani Zhao, Cara Jade Myers, Keyla Monterroso Mejia, Samara Joy, Mason Thames, Park Ji-hu, Boman Martinez-Reid, Priya Kansara, Yasmin Finney, Bridgette Doremus, Aria Mia Loberti, Isabel Gravitt, Delaney Rowe, Armen Nahapetian, Aditya Kusupati, Vedang Raina, Arsema Thomas, Adwa Bader, Amaury Lorenzo, Corey Mylchreest, Sam Nivola, Gabby Windey, Cwaayal Singh, Jaylin Webb, Kudakwashe Rutendo, Chintan Rachchh, Sajith Rajapaksa, Diego Calva, Pardis Saremi, Dominic Sessa, India Amarteifio, Mia Challiner, Aryan Simhadri

## M.2 Politicians

We collect politicians from Wikipedia [58]. The distribution of the caption counts of the sampled politicians is given in Table 7. The sampled politicians in the descending order of their number of caption counts are:

Barack Obama, John Lewis, Theresa May, Narendra Modi, Kim Jong-un, David Cameron, Angela Merkel, Bill Clinton, Xi Jinping, Justin Trudeau, Emmanuel Macron, Nancy Pelosi, Arnold Schwarzenegger, Ron Paul, Shinzo Abe, Adolf Hitler, John Paul II, Tony Blair, Sachin Tendulkar, Nick Clegg, Newt Gingrich, Scott Morrison, Arvind Kejriwal, Ilham Aliyev, Jacob Zuma, Bashar al-Assad, Laura Bush, Sonia Gandhi, Kim Jong-il, Robert Mugabe, James Comey, Rodrigo Duterte, Pete Buttigieg, Lindsey Graham, Hosni Mubarak, Enda Kenny, Alexei Navalny, Rob Ford, Leo Varadkar, Evo Morales, Lee Hsien Loong, Henry Kissinger, Petro Poroshenko, Joko Widodo, Clarence Thomas, Mohamed Morsi, Mahathir Mohamad, Juan Manuel Santos, Abiy Ahmed, William Viktor Orban, Uhuru Kenyatta, Mike Huckabee, Sheikh Hasina, Martin Schulz, Giuseppe Conte, John Howard, Benito Mussolini, Tulsi Gabbard, Dominic Raab, Michael D. Higgins, François Hollande, Yasser Arafat, Mark Rutte, Mahathir Mohamad, Juan Manuel Santos, Abiy Ahmed, William Prince, Lee Kuan Yew, Mikhail Gorbachev, Hun Sen, Jacques Chirac, Martin O'Malley, Benazir Bhutto, Yoshihide Suga, John Major, Muammar Gaddafi, Jerry Springer, Sandra Day O'Connor, Madeleine Albright, Thomas Mann, Paul Kagame, Simon Coveney, Grant Shapps, Sebastian Coe, Merrick Garland, Jean-Yves Le Drian, Nursultan Nazarbayev, Horst Seehofer, Liz Truss, Rowan Williams, Ellen Johnson Sirleaf, George Weah, Mark Sanford, Yoweri Museveni, Luigi Di Maio, Ben Wallace, Herman Van Rompuy, Daniel Ortega, Olaf Scholz, Beppe Grillo, Alassane Ouattara, Nicolás Maduro, Tamim bin Hamad Al Thani, Mary McAleese, Asif Ali Zardari, Joseph Goebbels, Nikol Pashinyan, Deb Haaland, Paul Biya, Abdel Fattah el-Sisi, Thabo Mbeki, Kyriakos Mitsotakis, Joseph Muscat, Micheál Martin, Rebecca Long-Bailey, Paschal Donohoe, Todd Young, Jean-Marie Le Pen, Nick Griffin, Zoran Zaev, Pierre Nkurunziza, Abhisit Vejjajiva, Maggie Hassan, Steven Chu, Juan Guaidó, Edi Rama, Mary Landrieu, Jyrki Katainen, Jens Spahn, John Dramani Mahama, Gina Raimondo, Alec Douglas-Home, Viktor Orbán, Anita Anand, Isaias Afwerki, James Cleverly, Ibrahim Mohamed Solih, Leymah Gbowee, Václav Havel, John Rawls, Jack McConnell, Romano Prodi, Eoghan Murphy, Vicky Leandros, Norodom Sihamoni, Nayib Bukele, Shirin Ebadi, Jusuf Kalla, George Eustice, Joachim von Ribbentrop, Peter Altmaier, Akbar Hashemi Rafsanjani, Paul Singer, Christian Stock, Moussa Faki, Dominique de Villepin, Michael Fabricant, Kim Dae-jung, Eamon Ryan, Shavkat Mirziyoyev, Denis Sassou-Nguesso, Werner Faymann, Kamla Persad-Bissessar, Ingrid Betancourt, Volodymyr Zelenskyy, Park Chung Hee, Elvira Nabiullina, Roselyne Bachelot, Heinz Fischer, Hideki Tojo, Anatoly Karpov, Marcelo Ebrard, Slavoj Žižek, Trent Lott, Alfred Rosenberg, Gabi Ashkenazi, Valentina Matviyenko, Kgalema Motlanthe, Pedro Castillo, Winona LaDuke, Peter Bell, Boyko Borisov, Carl Bildt, Almazbek Atambayev, Andry Rajoelina, Carl Schmitt, Ralph Gonsalves, Liam Byrne, Alok Sharma, Jean-Michel Blanquer, Robert Schuman, Shinzō Abe, Doris Leuthard, Jacques Delors, Floella Benjamin, Sauli Niinistö, Annalena Baerbock, Toomas Hendrik Ilves, Alejandro Giammattei, Bob Kerrey, Timothy McCully, Stefan Löfven, Javier Solana, Salva Kiir Mayardit, Cecil Williams, Shahbaz Bhatti, Marianne Thyssen, Marty Natalegawa, Roh Moo-hyun, John Diefenbaker, Antonio Inoki, Iván Duque, CY Leung, Tom Tancredo, Sigrid Kaag, Jim Bolger, Lou Barletta, Li Peng, Laura Chinchilla, Gennady Zyuganov, Chen Shui-bian, Sebastián Piñera, Gustavo Petro, Miguel Díaz-Canel, Alberto Fernández, Gerald Darmanin, Boutros Boutros-Ghali, Joschka Fischer, Maia Sandu, Ricardo Martinelli, Andrej Babiš, Dan Jarvis, Nikos Dendias, Chris Hipkins, Tawakkol Karman, Booth Gardner, Karin Kneissl, Mobutu Sese Seko, Alexander Haig, Alexander De Croo, Ahmed Aboul Gheit, Yasuo Fukuda, Jean-Luc Mélenchon, Jane Ellison, Diane Dodds, Helen Whately, Idriss Déby, Patrice Talon, Carmen Calvo, Dario Franceschini, Emma Bonino, Richard Ferrand, Andreas Scheuer, Moshe Katsav, K. Chandrashekar Rao, P. Harrison, Robert Habeck, Ann Linde, Jon Ashworth, Edward Scicluna, Stef Blok, Lawrence Gonzi, William Roper, Josep Rull, Sam Kutesa, Raja Pervaiz Ashraf, David

936 ↪ Cairns, Ilir Meta, Perry Christie, Rinat Akhmetov, Ahmet Davutoğlu, Franck Riester, Nikos Christodoulides, Damien O'Connor, Sali Berisha,
937 ↪ Umberto Bossi, Lee Cheuk-yan, Alpha Condé, Alexander Newman, Annette Schavan, Yuri Andropov, Peter Tauber, Faure Gnassingbé, Bolkiah of Brunei,
938 ↪ Karl-Theodor zu Guttenberg, Michael Brand, Helen Suzman, Ron Huldai, Mohamed Azmin Ali, François-Philippe Champagne, Agostinho Neto, Marielle
939 ↪ de Sarnez, Kurt Waldheim, Mounir Mahjoubi, Juan Orlando Hernández, Angela Kane, Lech Wałęsa, Luis Lacalle Pou, Barbara Pompili, Margaritis
940 ↪ Schinas, Tigran Sargsyan, Wolfgang Bosbach, Raed Saleh, Johanna Wanka, Michelle Donelan, Roberto Speranza, Traian Băsescu, Iurie Leancă, Dara
941 ↪ Calleary, Ilona Staller, Micheline Calmy-Rey, Thomas Oppermann, Karine Jean-Pierre, Luciana Lamorgese, Azali Assoumani, Michael Adam, Paulo
942 ↪ Portas, Svenja Schulze, Pita Sharples, Choummaly Sayasone, Federico Franco, Félix Tshisekedi, Roberta Metsola, Nia Griffith, Paul Myners, Ahmad
943 ↪ Vahidi, Kaja Kallas, Hua Guofeng, Olga Rypakova, Otto Grotewohl, Audrey Tang, Oskar Lafontaine, Ivica Dačić, Isa Mustafa, Xiomara Castro, M. G.
944 ↪ Ramachandran, Fernando Grande-Marlaska, Wopke Hoekstra, Tomáš Petříček, Egils Levits, Roland Koch, Joseph Deiss, Laurentino Cortizo, Alan
945 ↪ García, Nikola Poposki, Evarist Bartolo, Reyes Maroto, Zuzana Čaputová, Sergei Stanishev, Plamen Oresharski, Ana Brnabić, Carlos Alvarado
946 ↪ Quesada, Marek Biernacki, Olivier Véran, Vjekoslav Bevanda, Clare Moody, Matthias Groote, Giorgos Stathakis, Marta Cartabia, Elena Bonetti,
947 ↪ Dina Boluarte, Milo Đukanović, Levan Kobiashvili, Isabel Celaá, Jarosław Gowin, José Luis Escrivá, Cora van Nieuwenhuizen, Ivan Mikloš, Arancha
948 ↪ González Laya, Viola Amherd, Gernot Blümel, José Luis Ábalos, Deo Debattista, Alain Krivine, Zlatko Lagumdžija, Edward Argar, Adrian Năstase,
949 ↪ Zdravko Počivalšek, Miroslav Kalousek, Gabriel Boric, Juan Carlos Campo, Karel Havlíček, Kiril Petkov, Elżbieta Rafalska, Tobias Billström,
950 ↪ Miroslav Toman, Mihai Răzvan Ungureanu, Ivaylo Kalfin, Élisabeth Borne, Herbert Fux, Petru Movilă, Koichi Tani, Caroline Edelstam, Barbara
951 ↪ Gysi, Ľubomír Jahnátek, Nuno Magalhães, Martin Pecina, Goran Knežević, Björn Böhning, Iñigo Méndez de Vigo, Božo Petrov, Ian Karan, Hernando
952 ↪ Cevallos, Milan Kujundžić, Adriana Dăneasă, Ida Karkiainen, Zoran Stanković, Boris Tučić, Jerzy Kropiwnicki, Rafael Català Polo, Ljube
953 ↪ Boškoski, Camelia Bogdănici, Józef Oleksy, Frederik François, Zbigniew Ćwiąkalski, Herbert Bösch, Metin Feyzioğlu, Zoltán Illés, Vivi Friedgut

## M.3 Classical Artists

We collected classical artists from the https://www.wikiart.org, a website that collects various arts from different artists and categorizes them into pre-defined art style categories. For classical artists, we collected the artist names from the art styles: *Romanticism, Impressionism, Realism, Baroque, Neoclassicism, Rococo, Academic Art, Symbolism, Cubism, Naturalism.* The distribution of the caption counts of the sampled artists is given in Table 7. The sampled artists in the descending order of their number of caption counts are:

## M.4 Modern Artists

We also collected modern artists from the https://www.wikiart.org. For modern artists, we collected the artist names from the art styles: *Expressionism, Surrealism, Abstract Expressionism, Pop Art, Art Informel, Post-Painterly Abstraction, Neo-Expressionism, Post-Minimalism, Neo-Impressionism, Neo-Romanticism, Post-Impressionism.* The distribution of the caption counts of the sampled artists is given in Table 7. The sampled artists in the descending order of their number of caption counts are:

```
1021      Vincent Van Gogh, David Bowie, Andy Warhol, Pablo Picasso, Frida Kahlo, Keith Haring, Salvador Dali, Paul Gauguin, Camille Pissarro, Paul
1022  ↪   Cezanne, Henri Matisse, Paul Klee, Francis Bacon, Edvard Munch, Amedeo Modigliani, Egon Schiele, Jean-Michel Basquiat, David Lynch, Wassily
1023  ↪   Kandinsky, Peter Max, Roy Lichtenstein, Paul Reed, Franz Marc, David Smith, Mark Rothko, Georges Seurat, Leroy Neiman, Joan Miro, Jackson
1024  ↪   Pollock, August Macke, Man Ray, Piet Mondrian, Cy Twombly, Henri De Toulouse-Lautrec, Graham Bell, Paul Signac, Robert Indiana, Yayoi Kusama,
1025  ↪   Rene Magritte, Jasper Johns, Walter Crane, Robert Morris, Emily Carr, Lucian Freud, Ernst Ludwig Kirchner, Tom Thomson, Anish Kapoor, Alex
1026  ↪   Katz, Pierre Bonnard, John Cage, Jim Dine, Ellsworth Kelly, Peter Blake, William Scott, Erin Hanson, Marcel Duchamp, Frank Stella, Robert
1027  ↪   Motherwell, Max Weber, Louise Nevelson, Peter Phillips, Willem De Kooning, Corneille, Wayne Thiebaud, Joan Mitchell, Jean Cocteau, Raoul Dufy,
1028  ↪   Antony Gormley, Max Ernst, Alberto Giacometti, Vanessa Bell, Richard Diebenkorn, James Rosenquist, Edouard Vuillard, Richard Hamilton, M.C.
1029  ↪   Escher, Sam Francis, Sean Scully, Anselm Kiefer, Edward Weston, Karel Appel, Philip Guston, Julian Schnabel, Ray Parker, James Ensor, Balthus,
1030  ↪   George Segal, Francis Picabia, Emil Nolde, Georges Rouault, Alice Neel, Helen Frankenthaler, Claes Oldenburg, Theo Van Rysselberghe, Maya Lin,
1031  ↪   Maurice Utrillo, Eric Fischl, H.R. Giger, Maurice Denis, Friedensreich Hundertwasser, Will Barnet, Paula Modersohn-Becker, Suzanne Valadon,
1032  ↪   Bruce Nauman, El Anatsui, Lee Krasner, Joseph Cornell, Patrick Heron, James Brooks, Paula Rego, Paul Jenkins, Jules Pascin, Lyonel Feininger,
1033  ↪   Edward Ruscha, Norman Lewis, Barnett Newman, David Park, Marie Laurencin, Rufino Tamayo, Chris Ofili, Lynd Ward, Jean-Paul Riopelle, Roger Fry,
1034  ↪   Remedios Varo, Maxime Maufra, Paul Serusier, Jacob Epstein, Richard Deacon, Walter Sickert, Mark Tobey, Jan Toorop, Jacek Yerka, Red Grooms, Ad
1035  ↪   Reinhardt, Eva Hesse, Oskar Kokoschka, Michael Sowa, Jean David, Sam Gilliam, Phyllida Barlow, Howard Finster, Augustus John, Elaine De
1036  ↪   Kooning, Beauford Delaney, David Hammons, Erich Heckel, Amrita Sher-Gil, Arthur Lismer, Mona Hatoum, Etel Adnan, Brion Gysin, John Chamberlain,
1037  ↪   Corita Kent, Allen Jones, Asger Jorn, Martin Kippenberger, George Tooker, Desmond Morris, Wolf Kahn, Jay Defeo, Irma Stern, Walasse Ting, Emile
1038  ↪   Bernard, Kathe Kollwitz, Frank Bowling, John Heartfield, Auguste Herbin, Frances Hodgkins, Meret Oppenheim, Andre Masson, Karl
1039  ↪   Schmidt-Rottluff, Hans Bellmer, Marino Marini, Morris Louis, Pyotr Konchalovsky, Richard Artschwager, Louis Cane, Betty Parsons, Max Pechstein,
1040  ↪   Richard Pousette-Dart, Georges Lemmen, Cuno Amiet, Louis Valtat, Kit Williams, Grace Cossington Smith, John Hoyland, Dennis Oppenheim, Lynda
1041  ↪   Benglis, James Lee Byars, Boris Grigoriev, Lili Elbe, Victor Brauner, Adrian Ghenie, Gillian Ayres, Ossip Zadkine, Alice Bailly, Felix
1042  ↪   Gonzalez-Torres, Johannes Itten, Charles Long, John Marin, Winifred Nicholson, Alfred Kubin, Charles Angrand, Zinaida Serebriakova, John Duncan
1043  ↪   Fergusson, Norman Bluhm,
1044      Harald Sohlberg, Zdzislaw Beksinski, Barkley L. Hendricks, Bruno Schulz, Toyen, Pierre Alechinsky, Hippolyte Petitjean, Nicolas De Staël,
1045  ↪   Rainer Fetting, Hiro Yamagata, Lorser Feitelson, Taro Yamamoto, Kazuo Shiraga, Alberto Burri, Anne Truitt, Jozsef Rippl-Ronai, Ronald Davis,
1046  ↪   Tsuguharu Foujita, Wols, Keith Sonnier, Henry Van De Velde, Chang Dai-Chien, Stanley Whitney, Ann Hamilton, John Brack, Jules-Alexandre Grun,
1047  ↪   Billy Apple, Eileen Agar, Benny Andrews, Moise Kisling, Edward Wadsworth, Paul Thek, Audrey Flack, Allan D'Arcangelo, Rik Wouters, Charles
1048  ↪   Cottet, Gene Davis, Prudence Heward, Alexander Liberman, David Batchelor, Tadanori Yokoo, Frederick Sommer, Hedda Sterne, Othon Friesz, Roderic
1049  ↪   O'Conor, Santiago Rusinol, Richard Gerstl, Marianne Von Werefkin, Octavio Ocampo, Kay Sage, Jessica Stockholder, Gabriele Munter, Jean Benoit,
1050  ↪   May Wilson, Jean Paul Lemieux, Jack Tworkov, Abraham Manievich, Perle Fine, Renato Guttuso, Al Held, Martial Raysse, Le Pho, Charles Reiffel,
1051  ↪   Bernard Cohen, Rosalyn Drexler, Ilya Mashkov, Jack Youngerman, Ernst Wilhelm Nay, Adja Yunkers, Leon Spilliaert, Valerio Adami, Karl Benjamin,
1052  ↪   Luigi Serafini, Leon Polk Smith, Oswaldo Guayasamin, Blinky Palermo, Ferdinand Du Puigaudeau, Esteban Vicente, Matsutani, Oscar Dominguez,
1053  ↪   Yasuo Kuniyoshi, Sergei Parajanov, Joy Hester, Forrest Bess, Taro Okamoto, Maurice Tabard, Yiannis Moralis, Igor Grabar, Alex Hay, Albert
1054  ↪   Irvin, Amadeo De Souza-Cardoso, Robert Swain, Bradley Walker Tomlin, Kishio Suga, Carlos Almaraz, Manuel Alvarez Bravo, Dan Christensen, Cyril
1055  ↪   Power, Marcel Barbeau, Jeremy Moon, Jorge Castillo, Josef Capek, Maggie Laubser, John Altoon, Albert Dubois-Pillet, William Baziotes, Joseph
1056  ↪   Marioni, Michael Hafftka, Raoul Ubac, Tony Feher, Walter Battiss, Friedel Dzubas, Varlin, Alfred Manessier, Ron Gorchov, Tony Scherman, Alina
1057  ↪   Szapocznikow, Nikolaos Lytras, Carl Holsøe, Constantin Brâncuşi, Walter Osborne, Max Kurzweil, Jose Guerrero, Leon Underwood, Istvan Nagy,
1058  ↪   Albert Bloch, Ward Jackson, Piero Dorazio, Giorgio Griffa, Lourdes Castro, Lita Albuquerque, Thomas Downing, Pierre Tal-Coat, Mario Prassinos,
1059  ↪   Panayiotis Tetsis, Robert Goodnough, Paul Feeley, Michel Majerus, Marc Vaux, Konstantinos Maleas, Vladimir Dimitrov, Meijer De Haan, Guido
1060  ↪   Molinari, Arthur Beecher Carles, Bertalan Por, Christo Coetzee, Jammie Holmes, Lasar Segall, Enrico Donati, Jerzy Nowosielski, Gianfranco
1061  ↪   Baruchello, Luis Feito, Burhan Dogancay, Iosif Iser, Charles Gibbons, Thalia Flora-Karavia, Aldo Mondino, Pierre Daura, Josef Sima, Nikola
1062  ↪   Tanev, Konrad Klapheck, Theophrastos Triantafyllidis, Edvard Weie, Gerard Fromanger, Matthias Laurenz Gräff, Victor Servranckx, Istvan Farkas,
1063  ↪   Ramon Oviedo, Manabu Mabe, Grégoire Michonze, Stanisław Ignacy Witkiewicz, Abidin Dino, Esteban Frances, Alberto Sughi, Olga Albizu, Behjat
1064  ↪   Sadr, Jose De Guimaraes, Robert Nickle, Dale Hickey, Inigo Manglano-Ovalle, Antonio Carneiro, Horia Damian, Jacqueline Hick, Kuno Gonschior,
1065  ↪   Huguette Arthur Bertrand, Ethel Léontine Gabain, Helen Dahm, Ion Nicodim, Lucy Ivanova, Gil Teixeira Lopes, Michel Carrade, Florin Maxa,
1066  ↪   Jean-Paul Jerome, Vangel Naumovski, Graca Morais, Antonio Areal, Petros Malayan, Rodolfo Arico, Stefan Sevastre, Johannes Sveinsson Kjarval,
1067  ↪   Ilka Gedo, Lucia Demetriade Balacescu, Natalia Dumitresco, Rene Bertholo, Vasile Kazar, Petre Abrudan, Aurel Cojan, Tia Peltz, Alvaro Lapa
```

## N   Compute Used

We use 8 L40 GPUs to generate images for the all text-to-image models in our work. Overall, we use them for 16 hours per prompt, per dataset, per model to generate images. We downloaded the images on the same machine using 40 CPU cores, a process that took about 8 hours per dataset. For generating the image embeddings, we use the same 8 L40 GPUs, a process that took about 16 hours per dataset. The computation of imitation score and plotting are done on single CPU core on the same machine, a process that takes less than 30 minutes per dataset.

