# OpenReview forum: "How Many Van Goghs Does It Take to Van Gogh? Finding the Imitation Threshold"
_NeurIPS.cc/2024/Workshop/SafeGenAi — SafeGenAi Poster_

### Official Review · Reviewer_xS21 · 2024-10-08
**valuable topic yet the conclusion is inconvincing**

**Rating:** 6
**Confidence:** 4

**Review:**

paper summary:

the authors proposed the FIT problem to explore how many training images is enough to imitate a concept. Experiments reveal that the
 200-600 images could be the threshold. Experiments are conducted on face datasets and art style datasets. This problem is valuable to  detecting copyright  violations.

pros:

 1. experiments are concrete.
2. The topic is of great value to copyright and privacy in image generation.

cons:

while the authors provide concrete experiments, I  doubt the correctness of conclusion. In the domain of text-guided image editing, a SOTA method called Imagic in CVPR 2023, and its speedup and less overfitting version, Forgedit, have demonstrated that with only one image, the diffusion model is almost capable of learning every details of the reference image. I would like to see the authors  compare with these two methods to demonstrate why one image is not enough to 'make it Van Gogh' if this paper is finally accepted.

---

### Official Review · Reviewer_5Jpn · 2024-10-09
**The paper's main contribution lies around finding frequency of target domain in training images (concept frequency), similarity of generated images (concept similarities) and detecting the imitation threshold using change detection algorithm**

**Rating:** 6
**Confidence:** 4

**Review:**

The paper's main contribution lies around finding frequency of target domain in training images (concept frequency), similarity of generated images (concept similarities) and detecting the imitation threshold using change detection algorithm
Pros
- The paper presents a well-structured approach with clear explanations of the steps involved in estimating the imitation threshold (FIT). It demonstrates an understanding of the cost constraints associated with retraining models from scratch, offering a more efficient alternative through MIMETIC2.
- The introduction defines the problem of imitation well, and the background section contextualizes the current research landscape, allowing readers to follow the paper without confusion.
-  The work is significant because it highlights the practical consequences of text-to-image models in terms of ethical and legal violations. The imitation threshold could serve as a basis for future debates on model transparency and responsibility.


Cons
- The authors concluded
> models can't replicate concepts that appear less than 200 times

While this is sound theoretically based on the hypothesis, the authors should validate this through pretraining models and perform empirical analysis
- The proposed methodology (MIMETIC2) builds on existing methods of evaluating similarity in images, and while useful, it doesn’t introduce radically new concepts to the field
- Although the results are clear, the broader implications of these findings for legal or technical communities are not fully explained. For instance, the paper mentions copyright infringement potential but doesn’t clarify how the thresholds it proposes might help in setting legal standards.